# MYC-driven epigenetic reprogramming favors the onset of tumorigenesis by inducing a stem cell-like state

Vittoria Poli[1,2], Luca Fagnocchi[1,2,3], Alessandra Fasciani[1,2], Alessandro Cherubini[2], Stefania Mazzoleni [2], Sara Ferrillo[2], Annarita Miluzio[2], Gabriella Gaudioso[2,3,4], Valentina Vaira[2,3,4], Alice Turdo[5], Miriam Gaggianesi[5], Aurora Chinnici[5], Elisa Lipari[5], Silvio Bicciato[6], Silvano Bosari[3,4], Matilde Todaro [7] & Alessio Zippo[1,2,3]

Breast cancer consists of highly heterogeneous tumors, whose cell of origin and driver oncogenes are difficult to be uniquely defined. Here we report that MYC acts as tumor reprogramming factor in mammary epithelial cells by inducing an alternative epigenetic program, which triggers loss of cell identity and activation of oncogenic pathways. Over-expression of MYC induces transcriptional repression of lineage-specifying transcription factors, causing decommissioning of luminal-specific enhancers. MYC-driven dedifferentiation supports the onset of a stem cell-like state by inducing the activation of de novo enhancers, which drive the transcriptional activation of oncogenic pathways. Furthermore, we demonstrate that the MYC-driven epigenetic reprogramming favors the formation and maintenance of tumor-initiating cells endowed with metastatic capacity. This study supports the notion that MYC-driven tumor initiation relies on cell reprogramming, which is mediated by the activation of MYC-dependent oncogenic enhancers, thus establishing a therapeutic rational for treating basal-like breast cancers.

[1] Laboratory of Chromatin Biology & Epigenetics, Center for Integrative Biology (CIBIO), University of Trento, 38123 Trento, Italy. [2] Fondazione Istituto Nazionale di Genetica Molecolare "Romeo ed Enrica Invernizzi", Via F. Sforza 35, 20122 Milan, Italy. [3] Division of Pathology, Fondazione IRCCS Ca' Granda Ospedale Maggiore Policlinico, Milan 20122, Italy. [4] Department of Pathophysiology and Transplantation, University of Milan, Milan 20122, Italy. [5] Department of Surgical, Oncological and Stomatological Sciences, University of Palermo, Palermo 90127, Italy. [6] Center for Genome Research, University of Modena and Reggio Emilia, Modena 41125, Italy. [7] DiBiMIS, University of Palermo, Palermo 90127, Italy. These authors contributed equally: Vittoria Poli, Luca Fagnocchi. These authors jointly supervised this work: Matilde Todaro, Alessio Zippo. Correspondence and requests for materials should be addressed to A.Z. (email: alessio.zippo@unitn.it)

Tumorigenesis can be ascribed to a succession of genetic and epigenetic alterations that turn in heritable changes in gene expression programs, ultimately leading to the formation of a cell population characterized by functional and phenotypic heterogeneity[1,2]. Cell transformation frequently involves activation of developmental signaling programs, which endow cells with unlimited self-renewal potential and aberrant differentiation capability[3]. Somatic stem cells have been considered putative candidates for targets of transformation because of their inherent self-renewing capacity and their longevity, which would allow the acquisition of the combination of genetic and epigenetic aberrations sufficient for cell transformation[4]. Nevertheless, recent studies demonstrated that, upon oncogenic alterations, progenitors or committed cells can serve as tumor-initiating cells (TICs) by dedifferentiating and re-acquiring stem cell-like traits[5–7]. In the context of mammary gland tumorigenesis, it has been demonstrated that the BRCA1 basal-like breast cancer subtype may arise from luminal progenitor cells[8,9]. More recently, it has been shown that expression of oncogenic PIK3CA$^{H1047R}$ in oncogene-driven normal lineage-restricted mouse mammary cells causes cell dedifferentiation and development of multi-lineage mammary tumors[10,11]. Although these findings highlighted a functional role for oncogene-driven cell dedifferentiation in tumor initiation, the molecular mechanisms underlying cell reprogramming are incompletely understood.

Cell reprogramming requires overcoming those epigenetic barriers that are involved in maintaining cell-specific transcriptional programs, thereby preserving cell identity[12–14]. The activation of a specific repertoire of cis-regulatory elements—enhancers—is critical for cell specification. Cooperative binding of lineage-determining (LDTF) and signal-dependent (SDTF) transcription factors dictates the spatio-temporal pattern of gene expression[15]. Enhancers are characterized by accessible chromatin, marked by the deposition of H3K4me1, and their activation is associated with an increment of H3K27 acetylation[16]. Given their pivotal role in the determination of cell identity, decommissioning of active enhancers represents a critical step towards cell reprogramming[17]. Of importance, evidence indicates that disregulation of chromatin players responsible for enhancer regulation could favor tumorigenesis by driving the aberrant activation of oncogenic transcriptional programs[18–22].

Among the transcription factors (TFs) with a documented function in somatic cell reprogramming[23], the proto-oncogene MYC has a pivotal role in growth control, differentiation, and apoptosis and its expression level is tightly regulated in physiological conditions[24]. In breast cancer, MYC deregulation has been associated with up to 40% of tumors, and its hyper activation is a hallmark of the basal-like subtype[24–26]. Despite MYC proven oncogenic potential and its known function in the maintenance of self-renewing capacity and pluripotency[27,28], a causal link between MYC role as reprogramming factor and its tumorigenic effects has not been investigated.

Here we demonstrate that MYC acts as an oncogenic reprogramming factor by inducing cell plasticity that predisposes mammary luminal epithelial cells to acquire basal/stem cell-like properties and to onset of tumorigenesis, giving rise to TICs endowed with long-term tumorigenic capacity and metastatic potential.

## Results

**MYC alters luminal epithelial cell identity**. In order to evaluate the role of MYC in perturbing cell identity of somatic cells, we transduced hTERT-immortalized human mammary epithelial cells (thereafter named IMEC) with a retroviral vector expressing low levels of the exogenous c-Myc (Fig. 1a). MYC overexpression

induced alteration of the epithelial morphology with cells loosing polarity and adhesion, growing in semi-adherent condition and forming spheroids (Fig. 1b). Importantly, upon MYC over-expression we observed a similar phenotype in the luminal breast cancer cell lines MCF7, T47D, and ZR751 (Supplementary Fig. 1 a, b). Of note, the observed phenotype could not solely rely on induction of EMT, as we did not detect induction of EMT-related TFs (Supplementary Fig. 1c). In addition, the modest level of MYC overexpression did not cause major changes in the cell cycle profile or timing of cell division in IMEC-MYC (Supplementary Fig. 1d, e).

Gene expression profiling showed that IMEC-WT and IMEC-MYC differ for the expression of a specific subset of genes (Supplementary Fig. 1f) whose regulations was not affected by global transcriptional amplification, as they showed equivalent total RNA/cell content and similar cell size (Supplementary Fig. 1g, h)[29,30]. Gene ontology (GO) analyses indicated that IMEC-MYC upregulated genes involved in metabolic processes and, at the same time, downregulated genes controlling developmental processes, cell adhesion, and extracellular matrix integrity (Supplementary Fig. 1i)[24].

To determine whether MYC-induced alterations at both the morphological and transcriptional level may trigger perturbation of cell identity, we compared the gene expression profile of IMEC WT and -MYC with gene expression signatures of mature (ML) and progenitor luminal (LP) cells. Gene set enrichment analysis (GSEA) revealed a marked downregulation of the ML program in cells overexpressing MYC, combined with a significant enrichment of the LP-specific signature (Fig. 1c and Supplementary Fig. 1j, k). Of importance, overexpression of MYC in luminal breast cancer cell lines also caused downregulation of ML-specific genes (Supplementary Fig. 2a). Hence, MYC overexpression in mammary luminal cells causes dedifferentiation towards a progenitor-like state.

**MYC downregulates the expression of ML-specific TFs**. On the basis of these results, we asked whether the expression of lineage-specific transcription factors (LSTFs) was perturbed in consequence of MYC overexpression. We found that IMEC-MYC downregulated ML-specific TFs while they did not show a global and consistent modulation of the expression level of LP-specific regulators (Fig. 1d)[31–33]. Importantly, genes whose expression is dependent on luminal-specific TFs binding on their cognate enhancers[34] were downregulated in IMEC-MYC (Fig. 1e). We therefore focused on GATA3 and ESR1 TFs, two master regulators of mammary gland morphogenesis and luminal cell differentiation[35,36]. We confirmed that their transcriptional downregulation was not restricted to IMEC (Fig. 1f) as the same pattern was induced by MYC overexpression in different luminal breast cancer cell lines (Supplementary Fig. 2b). Moreover, knocking-down the exogenous MYC was sufficient to revert the observed down-modulation of these master regulators of mammary epithelial cells (Supplementary Fig. 2c, e). We then asked whether GATA3 and ESR1 downregulation could be mediated by MYC binding to their cis-regulatory elements. Upon MYC overexpression, we measured a concomitant increase of MYC association and reduction of active histone marks on GATA3 and ESR1 regulatory elements (Fig. 1g). Considering that often MYC deregulation causes transcription repression of its targets by antagonizing the transcriptional activity of MIZ1[37,38], we determined whether the MYC-dependent downregulation of GATA3 and ESR1 could be mediated by MIZ1 binding. ChIP assay showed that MIZ1 associated on the analyzed cis-regulatory elements (Fig. 1g) and its knock-down reverted the MYC-driven

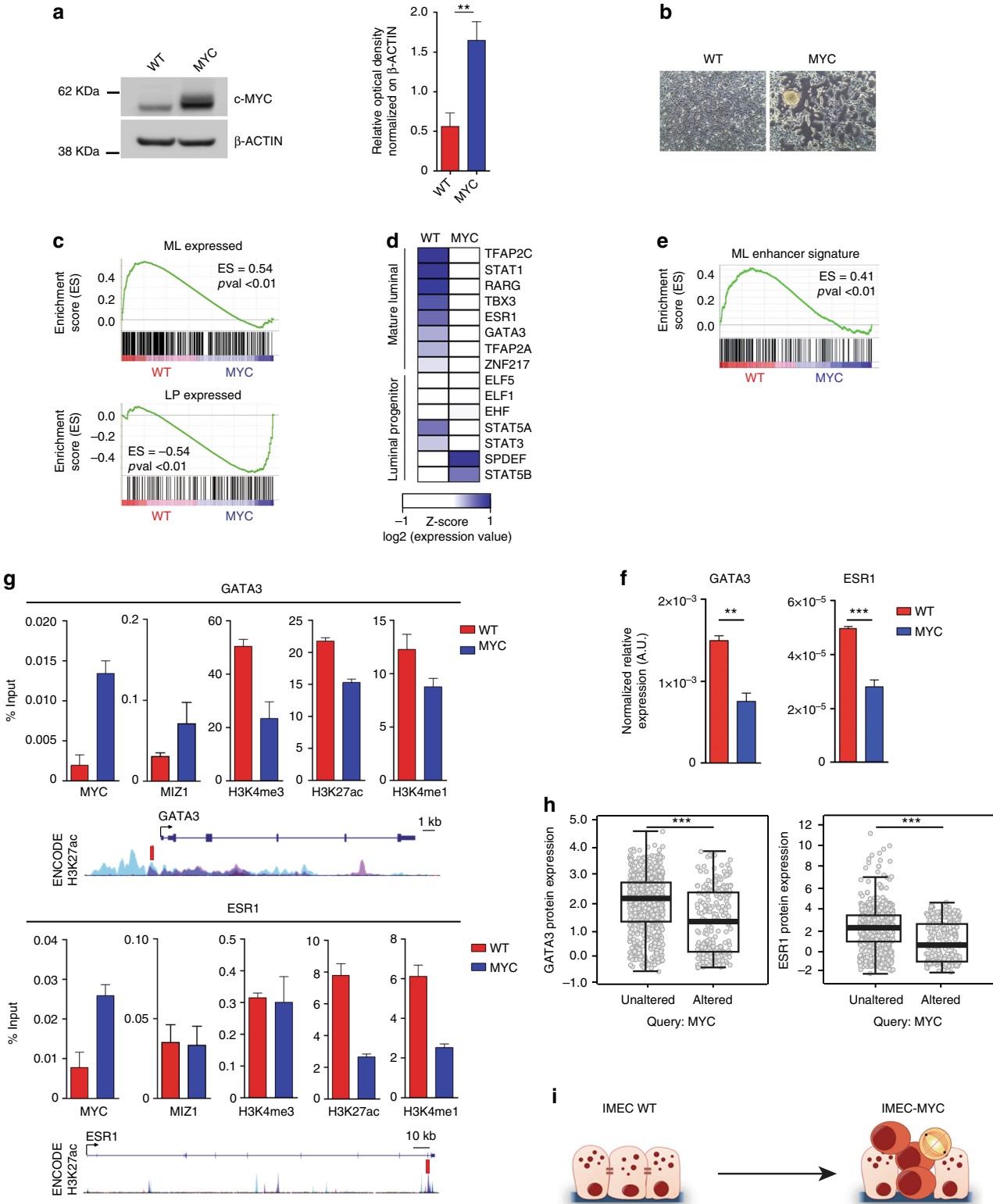

**Fig. 1** MYC inhibits the transcriptional program of mature luminal epithelial cells. **a** Western blot of c-MYC in IMEC WT and IMEC-MYC; β-ACTIN was used as loading control. Signal quantification is reported. Data are means ± SEM ($n = 3$). (**$P < 0.01$; Student's $t$-test). **b** Phase contrast images of confluent IMEC WT and IMEC-MYC. Scale bar, 200 μm. **c** GSEA of mature luminal (ML) and luminal progenitor (LP) gene signatures in IMEC WT vs. IMEC-MYC ($n = 3$). **d** Heatmap showing the expression of luminal specifying TFs in IMEC WT and IMEC-MYC. **e** GSEA of genes regulated by enhancers bound by mature luminal-specific TFs in IMEC WT vs. IMEC-MYC ($n = 3$). **f** qRT-PCR of GATA3 and ESR1 in IMEC WT and IMEC-MYC, normalized on spike-in RNAs. Data are means ± SEM ($n = 3$). (**$P < 0.01$, ***$P < 0.001$; Student's $t$-test). **g** ChIP-qPCR assessing MYC and MIZ1 binding and H3K4me3, H3K27ac, and H3K4me1 deposition at GATA3 promoter and ESR1 intronic enhancer in IMEC WT and IMEC-MYC. A scheme showing GATA3 and ESR1 PCR amplicons localization (red boxes) and layered H3K27ac signals from ENCODE is represented. Data are means ± SEM ($n = 3$). **h** Box plots representing the protein levels of GATA3 (left panel) and ESR1 (right panel) in breast cancer samples with unaltered or altered MYC level. **i** Scheme showing the MYC-induced reprogramming of IMEC WT to the luminal progenitor-like state of IMEC-MYC

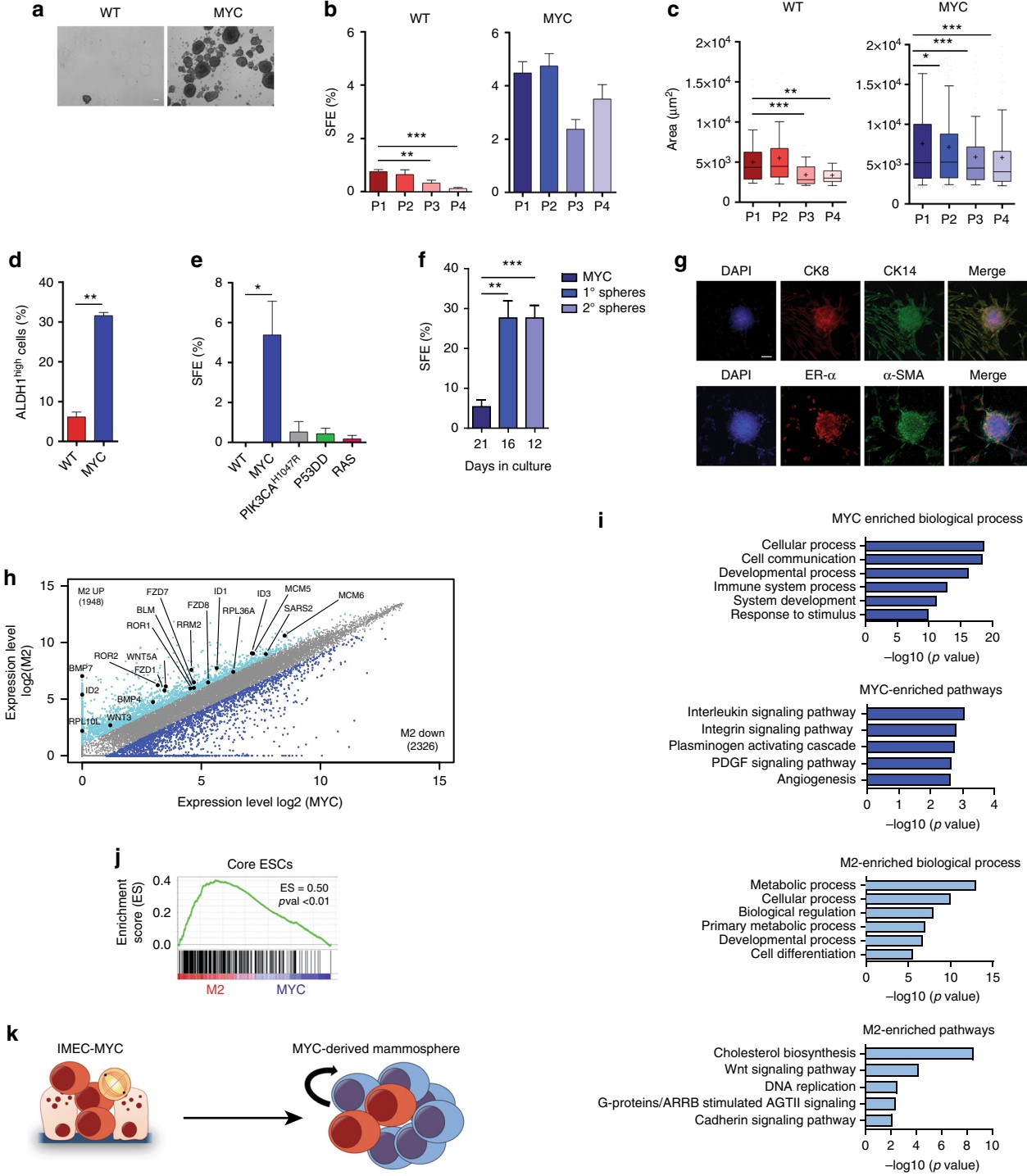

**Fig. 2** Sustained MYC overexpression confers stem cell-like traits. **a** Phase contrast images of IMEC WT and IMEC-MYC cultured in low adhesion conditions. Scale bar, 100 μm. **b** Spheres formation efficiency (SFE) of IMEC WT and IMEC-MYC at indicated passages ($n = 6$) (**$P < 0.01$, ***$P < 0.001$; Student's $t$-test). **c** Area (μm$^2$) of mammospheres formed by IMEC WT and IMEC-MYC at indicated passages ($n = 6$). Whiskers extend to 10th and 90th percentiles; central horizontal bar and black cross indicate median and mean, respectively (*$P < 0.05$, **$P < 0.01$, ***$P < 0.001$; Student's $t$-test). **d** Percentage of ALDH1-positive cells in IMEC WT and IMEC-MYC cultured in low adhesion conditions. Data are means ± SEM ($n = 3$) (**$P < 0.01$; Student's $t$-test). **e** Single cell SFE of IMEC WT, -MYC, -PIK3CA$^{H1047R}$, -P53DD and -RAS. Data are means ± SEM ($n = 3$) (*$P < 0.05$; Student's $t$-test). **f** Serial single-cell SFE of IMEC-MYC, IMEC-MYC-derived 1° Spheres (M1) and IMEC-MYC-derived 2° Spheres (M2). Single-cell-derived clones were obtained at the indicated time. Data are means ± SEM ($n = 4$) (**$P < 0.01$, ***$P < 0.001$; Student's $t$-test). **g** Immunofluorescence for basal (CK14 and α-SMA) and luminal (CK8 and ER-α) markers on differentiated M2. Scale bar, 50 μm. **h** Scatter-plot of gene expression profile of IMEC-MYC and M2. Genes up- (cyano) and downregulated (blue) in M2 with respect to IMEC-MYC are highlighted. Relevant M2 upregulated genes are indicated ($n = 3$). **i** GO analysis of differentially regulated genes between IMEC-MYC and M2 ($n = 3$). **j** GSEA of the core embryonic stem cell (ESCs) gene module in IMEC-MYC vs. M2 ($n = 3$). **k** Scheme representing IMEC-MYC and IMEC-MYC-derived mammospheres, enriched for cells with self-renewing capacity

transcription repression of GATA3 and ESR1 (Supplementary Fig. 2f, g).

To establish the pathological relevance of the anti-correlation between MYC overexpression and ESR1/GATA3 downregulation, we assessed the expression level of these ML-specific TFs in large cohorts of breast cancer samples. Analysis of different data sets of breast cancer patients[25,39] showed that MYC overexpression anti-correlated with ESR1 and GATA3 transcript levels (Supplementary Fig. 2h–j). Moreover, querying the proteome of genome-associated TCGA tumor samples[40] showed that the protein abundance of both ESR1 and GATA3 decreased in those breast cancers with augmented level of MYC (Fig. 1h). Together, these

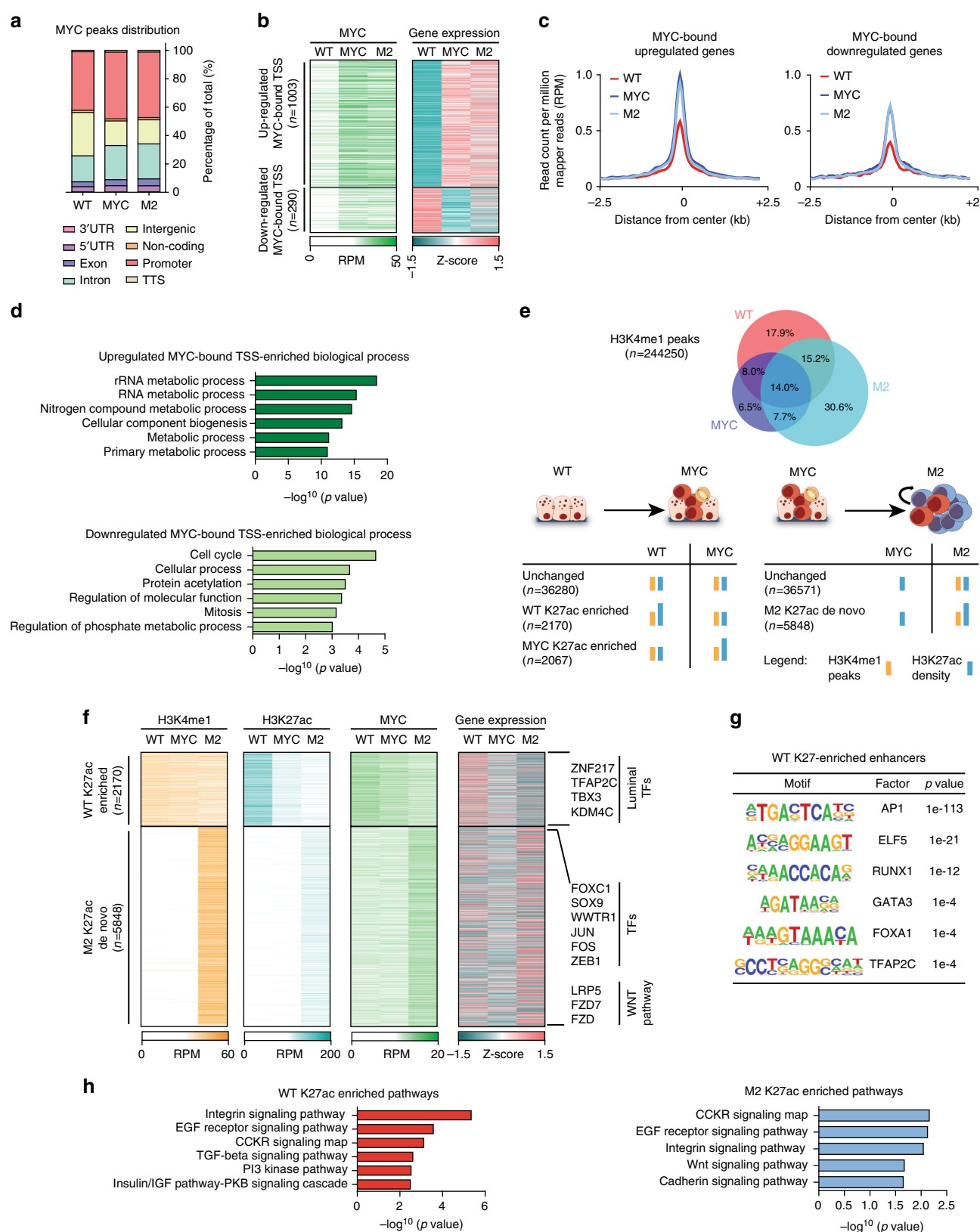

data indicate that MYC overexpression induced dedifferentiation of luminal cells by downregulating the expression of lineage-specific TFs, thereby supporting the reprogramming towards a progenitor-like state (Fig. 1i).

**Sustained MYC overexpression confers stem cell-like traits**. On the basis of the observed MYC-induced cell reprogramming, we asked whether MYC overexpression could enrich for cells with functional stem cell-like properties. We therefore measured the ability of IMEC WT and -MYC to grow for subsequent passages in low adherence conditions as mammospheres[41]. While WT cells formed mammospheres with low efficiency and did not proliferate beyond the second passage, cells overexpressing MYC showed enhanced sphere formation efficiency (SFE) (Fig. 2a–c). A similar increment in mammospheres formation was measured in luminal breast cancer cell lines upon MYC overexpression (Supplementary Fig. 3a, b). MYC sustained the propagation of mammospheres for several passages, indicating acquisition of long-term self-renewal capacity (Fig. 2b, c). Furthermore, IMEC-MYC mammospheres showed enrichment for cells expressing ALDH1, a distinctive marker of mammary stem cells[42] (Fig. 2d). Of importance the observed phenotype was a MYC-dependent effect, as IMEC expressing other oncogenic hits showed reduced long-term capacity to propagate as mammospheres (Supplementary Fig. 3c–f). To quantify the relative enrichment for cells endowed with self-renewing capacity, we performed single cell clonogenic assay. The obtained result indicated that IMEC WT could not give rise to any single cell-derived clone, while MYC overexpression was associated with the highest clonogenic potential (Fig. 2e). Moreover, single cell-derived primary spheres (named M1) were further enriched in cells with self-renewing capacity, showing higher SFE with respect to the parental heterogeneous population (Fig. 2f). Of importance, the measured enrichment of SFE was not due to clonal selection as independent single-cell isolated clones gave rise to similar increment in mammospheres formation (Supplementary Fig. 3g). Accordingly, knock-down of the exogenous MYC in mammospheres impaired the measured clonogenic potential (Supplementary Fig. 3h, i). Finally, we showed that under differentiation conditions, single cell-derived mammospheres expressed luminal (CK8 and ER-α) and myoepithelial (CK14 and α-SMA) markers, indicative of enrichment for stem cell-like cells endowed with multipotency (Fig. 2g).

To investigate whether MYC supported the activation of a stem cell-like transcriptional program, we profiled gene expression of single-cell-derived secondary mammospheres (clone M2 #1, thereafter named M2), determining differentially expressed genes with respect to IMEC-MYC (Fig. 2h). GO analyses showed that mammospheres were characterized by further upregulation of genes involved in metabolic pathways and downregulation of genes involved in developmental processes (Fig. 2h, i), indicative of reinforcement of MYC-driven transcriptional program. Furthermore, M2 upregulated genes involved in Wnt and Hippo

signaling pathways, which are critical regulators of stem cell self-renewing[43,44] (Fig. 2h, i and Supplementary Fig. 3j). Notably, GSEA analysis also revealed that the core embryonic stem cell-like gene signature was over-represented in M2 with respect to IMEC-MYC, and that genes codifying for MYC-related factors (MYC Module) significantly contributed to this transcriptional program (Fig. 2j and Supplementary Fig. 3k)[45]. Of note, comparative analyses of gene expression profiling showed that MYC induced a common transcriptional program among independent single-cell-derived clones of mammospheres (M2 clones #2−4) (Supplementary Fig. 4a, b). Moreover, the retrieved independent clones activated the same set of genes that were induced in M2 clone with respect to IMEC-MYC, including the embryonic stem cell-like gene signature (Supplementary Fig. 4c, e). Importantly, long-term maintenance of mammospheres and their subsequent sub-cloning was not supported by an increment of the MYC protein level and genomic instability (Supplementary Fig. 4f–h), thus arguing against clonal selection. Collectively, the above data suggest that MYC overexpression in luminal cells favor the onset of stem cell-like traits, such as sustained self-renewing capacity and re-activation of a pluripotency-associated transcriptional program (Fig. 2k).

**MYC induces an alternative epigenetic program**. To gain insights into the mechanisms through which MYC induces cellular reprogramming, we performed ChIP-seq analyses to profile chromatin modifications and the binding of MYC in IMEC WT, -MYC and mammospheres (Fig. 3). Considering that nearly 50% of MYC binding sites localized at promoters (Fig. 3a), we analyzed the transcriptional effects of increasing the MYC levels on these loci, in response to its overexpression. By ranking MYC-bound genes on the basis of their gene expression pattern, we defined two distinct subsets of targets whose expression augmented or decreased in response to MYC association, respectively (Fig. 3b and Supplementary Fig. 5a, b). Comparative analyses between these two subsets showed that in the steady state (IMEC WT) MYC occupancy was higher among the upregulated genes and it further increased in response to MYC overexpression (Fig. 3c and Supplementary Fig. 5c). Of note, the different MYC occupancy correlated with a specific enrichment for canonical E-box among the induced genes (Supplementary Fig. 5d), in agreement with previous reports[37,38]. Importantly, by analyzing previously published data sets[37,38], we confirmed that these two distinct subsets of MYC targets were induced and repressed accordingly, upon MYC activation in other two independent cell lines (Supplementary Fig. 5e, f). Considering that the transcriptional response to MYC overexpression has been correlated with the ratio between MYC and MIZ1 binding[37], we quantified their relative occupancy at the promoters of the two subsets (Supplementary Fig. 5g-j). We showed that in both analyzed data sets the downregulated genes had a lower MYC/MIZ1 ratio, supporting the notion that these targets are directly repressed by MYC in conjunction with MIZ1 binding, while high MYC/MIZ1 ratio

**Fig. 3** MYC induces an alternative epigenetic program in mammary epithelial cells. **a** Barplot showing the distribution of MYC peaks on indicated genomic features in IMEC WT, IMEC-MYC, and M2. **b** Heatmap showing the dynamic behavior of MYC normalized ChIP-seq signals on MYC-bound TSS. Expression of annotated genes is reported. RPM reads per million. **c** Tag density plots of MYC normalized ChIP-seq signals in IMEC WT, IMEC-MYC, and M2, centered on up- (left) and downregulated (right) TSS. RPM reads per million. **d** GO of up- (upper panel) and downregulated (lower panel) genes bound by MYC on their TSS. **e** Identification of modulated enhancers among IMEC WT, IMEC-MYC, and M2. In the upper panel, the Venn diagram shows the overlap of H3K4me1 ChIP-seq peaks among different cell types. In the lower panel, identified H3K4me1-positive regions were analyzed for their enrichment in H3K27ac, leading to identification of modulated and unchanged enhancers in the reported comparisons. **f** Heatmap showing the dynamic behavior of H3K4me1, H3K27ac, and MYC normalized ChIP-seq signals overidentified modulated enhancers. Expression of associated genes is reported. Key relevant genes associated to different enhancers groups are indicated on the right. RPM reads per million. **g** TF binding motifs enrichment at enhancers activated in IMEC WT. **h** GO of genes associated to differentially modulated enhancers in IMEC WT and M2

characterizes the upregulated genes, indicating a direct MYC-mediated transcriptional activation. Importantly, GO analyses showed that these two subsets were enriched for genes belonging to different functional categories (Fig. 3d). Supra-physiological expression of MYC has been associated to invasion of almost all active regulatory elements in the genome[29,30,37,46]. The specificity of the differential binding affinity and its association with transcription modulation was further supported by the ChIP-seq data analyses showing that more than 18,000 active promoters marked by H3K4me3 were not bound by MYC (Supplementary Fig. 6a). These results indicated that in this biological context MYC

activation did not caused chromatin invasion of active regulatory elements[29,30,37,46]. Together these analyses showed that MYC occupancy on promoters determined the transcriptional outcomes of MYC-target genes.

Considering that MYC also associated to introns and intergenic regions (Fig. 3a), we investigated whether it occupied and modulated the activation of enhancers. By profiling the distribution of H3K4me1 in IMEC WT, -MYC, and M2, we mapped all the putative distal *cis*-regulatory elements (Fig. 3e, upper panel). Thereafter, we defined the active enhancers by profiling the relative enrichment for H3K27ac at these loci. Overall, the cellular

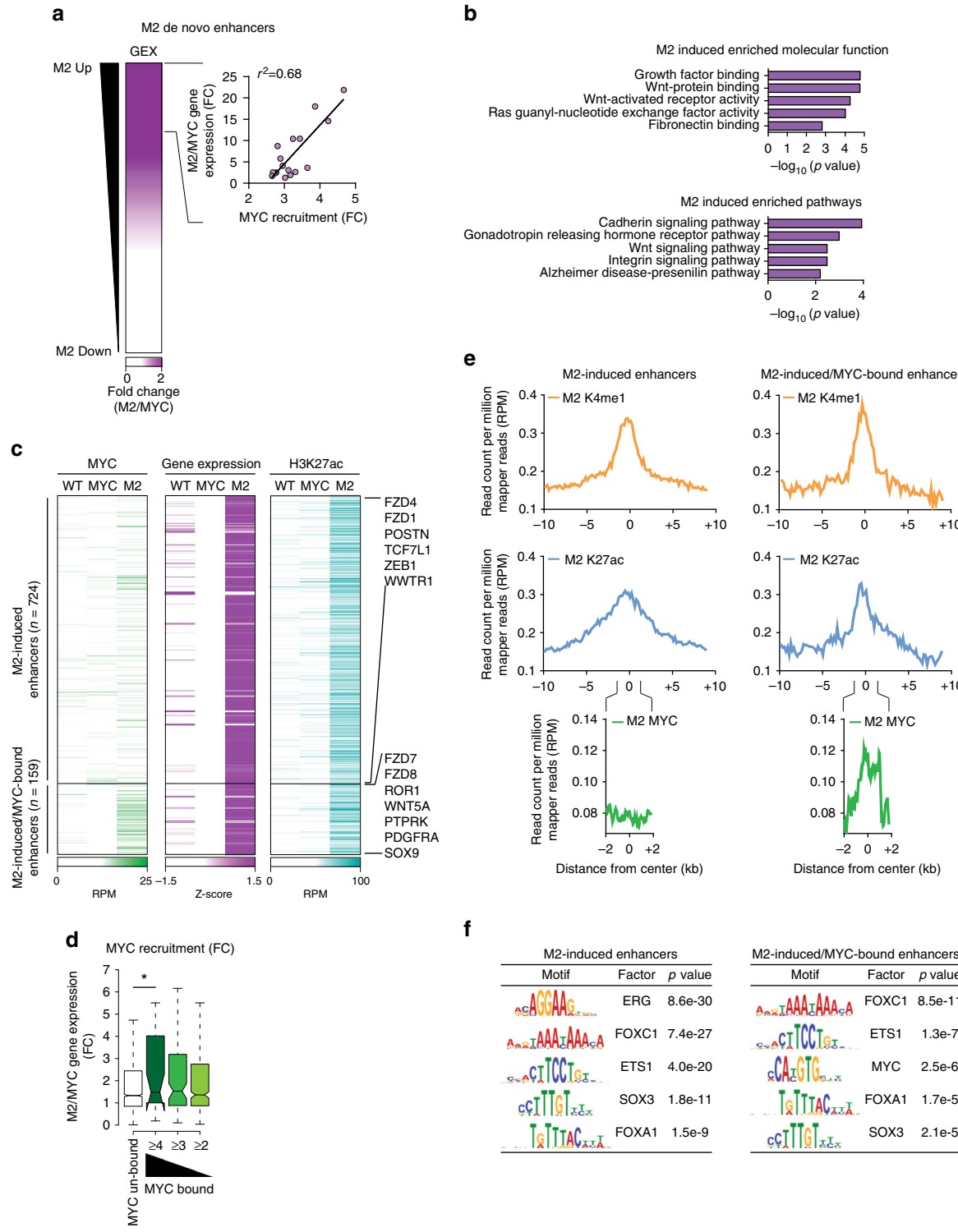

reprogramming was mirrored by a highly dynamic modulation of the defined *cis*-regulatory elements giving rise to different enhancer states (Fig. 3e, lower panel). The comparative analyses showed that a subset of enhancers were repressed in the MYC-overexpressing cells as they showed a consistent reduction of the H3K27ac level (Fig. 3f). Using a criterion of proximity to assign each enhancer to its regulated gene[47], we observed that enhancer decommissioning determined the downregulation of their related genes (Fig. 3f and Supplementary Fig. 6b). Among these genes we identified TFs involved in establishing the transcriptional regulatory network of luminal cells, such as TFAP2C, TBX3, and ZNF217 (Fig. 3f and Supplementary Fig. 6c). In addition, these repressed enhancers were enriched for binding sites of luminal-specific TFs (Fig. 3g), in accordance with the down-regulation of genes associated to ML-specific enhancers (Fig. 1e). Of note, GO analyses highlighted that the WT-specific activated enhancers were mainly related to genes involved in the integrin, EGF and PI3K signaling pathways (Fig. 3h).

By focusing on the chromatin modulations occurring in the mammospheres, we identified a subset of enhancers, which were specifically activated in M2 (Fig. 3e and Supplementary Table 1). These de novo enhancers were defined as distal genomic regions, which did not carry H3K4me1 and H3K27ac in IMEC and gained these histone modifications upon transition to a stem cell-like state (Fig. 3f). In addition, the activation of de novo enhancers was associated with an increment of MYC binding and with an overall increased expression of the related genes (Fig. 3f and Supplementary Fig. 6b, d, e). We found that stem cell-associated TFs and genes involved in activating the Wnt signaling were strongly enriched in this subset of enhancers (Fig. 3f–h). Taken together, these results indicated that the MYC-induced alteration of the luminal-specific transcriptional program associates with the repression of those enhancers that modulate the expression of the luminal lineage-specific TFs. In addition, the acquisition of a stem cell-like fate is associated with the activation of de novo enhancers that control the expression of TFs and signaling pathways which are frequently activated in both somatic and cancer stem cells[5,48–51].

**Activation of de novo enhancers drives oncogenic pathways.** We further characterized the de novo enhancer by ranking their related genes according to their expression level in mammospheres and we observed a positive correlation between over-expressed genes and increased MYC recruitment at their enhancers (Fig. 4a and Supplementary Fig. 7a). GO analyses showed that the enhancer-dependent regulated genes were associated with the modulation of Wnt pathways (Fig. 4b). Specifically, we identified genes coding for oncogenic TFs as well as genes involved in regulating both the canonical and non-canonical Wnt pathways, which are often deregulated in breast cancer (Fig. 4c and Supplementary Fig. 7b, c)[52–54]. By defining

the set of genes whose de novo enhancers were bound by MYC and induced in mammospheres, we showed that MYC associated with one third of the 289 regulated genes (Fig. 4c and Supplementary Fig. 7d). Of importance, the increment of expression of this subset of genes correlated with augmented MYC occupancy at the relative enhancers (Fig. 4d and Supplementary Fig. 7e, f). In addition the knock-down of the exogenous MYC, which cause a 50% reduction of total MYC protein (Supplementary Fig. 3h), impaired the transcriptional activation of its targets (Supplementary Fig. 8a). Next, we investigated the direct contribution of MYC binding to the chromatin state of the de novo enhancers by measuring the relative enrichment for H3K4me1, H3K27ac, and MYC at these loci (Fig. 4e). These analyses showed that the M2-induced enhancers are characterized by a large distribution of both H3K27 and K4me1 marks, spanning as average regions over 3.1 kb (Fig. 4e). Of note, the distribution of these histone marks is similar to the pattern of the stretch- and super-enhancers which compromise dense TFs binding sites, forming cluster of enhancers that regulate the expression of lineage-specifying genes[20,55]. In addition, on MYC-target de novo enhancers, we found that MYC binding peaked at the center of the H3K27ac-enriched region, suggesting a direct contribution to the deposition of this active histone mark (Fig. 4e). Moreover, by performing motif discovery analysis we found the highest enrichment for FOX- and SOX-family members, as well as ETS1 motifs (Fig. 4f). Importantly, among the MYC-target de novo enhancers, we found a specific enrichment for a non-canonical E-box[29], indicating that MYC association is mediated by its direct binding to the chromatin. In summary, these data strongly support the notion that de novo enhancers modulate the transcriptional activation of oncogenic pathways. In addition we characterized a subset of de novo enhancers, which are enriched for MYC binding at their epicenter, suggesting a modulatory function in their activation.

**WNT pathway supports MYC-induced stem cell features.** To establish whether this enhancer-mediated regulation determined the overall hyperactivation of the Wnt pathway in mammospheres, we verified the transcriptional upregulation of Wnt pathway-related genes, including the FZD1 and FZD8 receptors and LRP6 co-receptor (Fig. 5a). In addition, the two major inhibitors of the pathway, DKK1 and SFRP1, were strongly downregulated in cells overexpressing MYC (Fig. 5a). Next, to detect Wnt responsive cells, we transduced IMEC-MYC with a lentiviral vector containing a 7xTCF-eGFP reporter cassette (7TGP). FACS analyses showed that the Wnt pathway was activated in mammospheres and not in IMEC-MYC (Fig. 5b). In order to determine whether WNT signaling activation could have a functional role in MYC-induced stem cell features, we discerned between IMEC-MYC with the highest (GFP$^{high}$) and the lowest (GFP$^{low}$) signal for Wnt pathway activation. Dye retention assay showed that Wnt responsive cells (GFP$^{high}$) were enriched for

**Fig. 4** Activation of de novo enhancers drives oncogenic pathways. **a** On the left, heatmap showing the ranked differential expression profile of genes associated to M2 de novo enhancers in M2 vs. IMEC-MYC. On the right, correlation plot between the fold change (FC) expression and differential MYC binding in M2 vs. IMEC-MYC, on enhancers of induced genes in mammospheres. Each dot represents a bin of ten genes. **b** GO of genes associated to M2 de novo enhancers and transcriptionally induced in M2 vs IMEC-MYC. **c** Heatmap showing the dynamic behavior of MYC and H3K27ac normalized ChIP-seq signals overidentified enhancer regions whose associated genes are transcriptionally induced in M2 with respect to IMEC-MYC with either increased MYC binding in M2 or not. Expression of associated genes is reported. Relevant genes belonging to the two groups are indicated on the right. RPM reads per million. **d** Notched boxplot showing the distribution of the fold change (FC) of the expression values between M2 spheres and IMEC-MYC of genes associated to de novo enhancers differentially bound by MYC. Horizontal black lines indicate medians. Boxes extend from the first to third quartile and the Tukey method was used to plot whiskers. (*$P < 0.05$; Student's $t$-test). **e** Tag density plots of H3K4me1, H3K27ac, and MYC normalized ChIP-seq signals in IMEC WT, IMEC-MYC, and M2, centered on enhancers regions associated to genes which are either only transcriptionally induced (left) or also enriched for MYC binding (right) in M2 with respect to IMEC-MYC. **f** Tables depicting transcription factors binding motifs enrichment at the center (±2 kb) of enhancers associated to genes which are either only transcriptionally induced (left) or also enriched for MYC binding (right) in M2 with respect to IMEC-MYC

slow-dividing cells, which retained higher level of the cell tracer, suggesting enrichment for stem cells (Fig. 5c). Given the cellular heterogeneity within the mammospheres population, we performed single cell sorting of GFP$^{high}$ and GFP$^{low}$ cells (Fig. 5d and Supplementary Fig. 8c). On average, by analyzing independent clones we determined that the GFP$^{high}$ sub-population showed enrichment for cells with self-renewing capacity (Fig. 5e, left panel). We further characterized the GFP$^{high}$-derived primary spheres (GFP$^{high}$-derived M1) with respect to the relative enrichment for Wnt pathway activation. The obtained results

showed a concomitant increment of Wnt signaling in GFP$^{high}$-derived with respect to the GFP$^{low}$-derived M1 cells (Fig. 5e, left panel and Supplementary Fig. 8c). Furthermore, by performing serial clonogenic assay of both GFP$^{high}$ and GFP$^{low}$ cells derived from independent clones, we observed that the Wnt responsive population was further endowed with self-renewing capacity, giving rise to clones characterized by enhanced activation of the pathway (Fig. 5e, right panel and Supplementary Fig. 8c). Gene expression profiling of sorted GFP$^{high}$ and GFP$^{low}$ cells showed that Wnt-responsive cells were enriched for a mammary stem cell

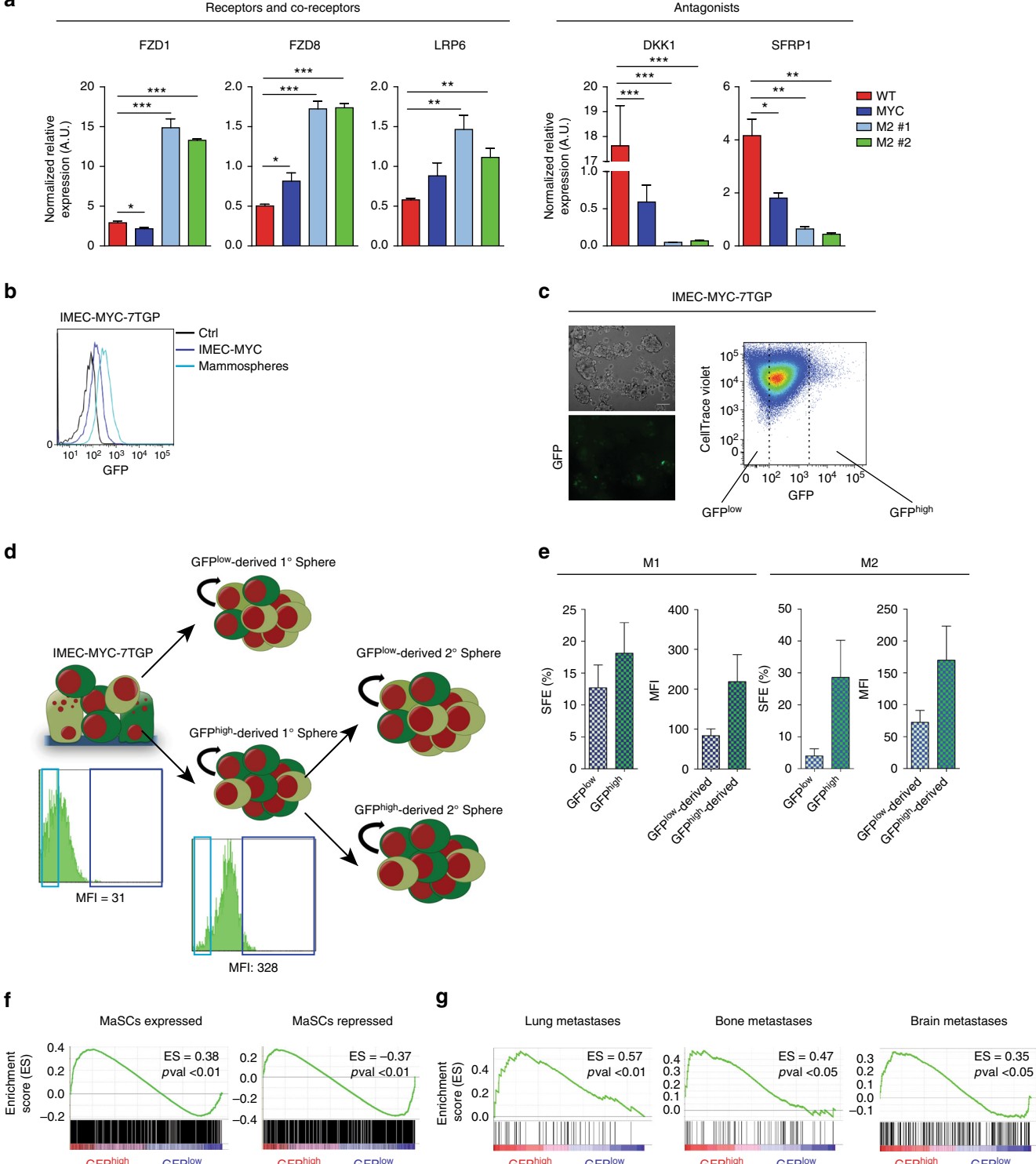

transcriptional program and correlated with metastatic transcriptional signatures (Fig. 5f, g). These results suggest a correlation between the reactivation of Wnt pathway and acquisition of a stem cell-like transcriptional program, which has been associated with increased risk of developing recurrent cancer[42,56,57].

**MYC-induced reprogramming favors the onset of TICs.** To determine whether MYC-induced reprogramming favors the onset of TICs in vivo, we challenged IMEC-MYC with an additional oncogenic insult by overexpressing PIK3CA[H1047R], which caused hyper-activation of PI3K pathway (Supplementary Fig. 9a). PIK3CA[H1047R] expression enhanced MYC-related phenotype and increased self-renewing capacity (Supplementary Fig. 8b, c). In addition, IMEC-MYC-PIK3CA[H1047R] cells formed about threefold more colonies in soft agar and showed migration capacity (Supplementary Fig. 9d, e), suggesting that they were enriched for transformed cells. To address this aspect, we injected IMEC-MYC-PIK3CA[H1047R] cells and the corresponding controls in the sub-renal capsule of immunocompromised mouse hosts. No tumors arose when an equal number of IMEC WT, -MYC or -PIK3CA[H1047R] were injected, which instead survived locally giving rise to distinguishable mammary gland-like structures (Fig. 6a). IHC analyses showed that transplantation of IMEC WT into a permissive in vivo context, which favors growth and formation of organized luminal-like structures[58,59], permitted the engraftment of these cells, maintaining features of luminal cells. Of note, MYC overexpression caused decrement of luminal markers CK8/18 and the downregulation of ER-α, thus recapitulating the transcriptional repression measured in vitro. In contrast, all mice injected with IMEC-MYC-PIK3CA[H1047R] cells formed tumors composed by highly proliferative (KI67[+]) and poorly dedifferentiated cells (Fig. 6a). By performing in vivo limiting dilution assay, we estimated a frequency of TICs of 1/832 (Supplementary Fig. 9f), suggesting that IMEC-MYC-PIK3CA[H1047R] were enriched for TICs. Of note, the obtained results were not dependent on the site of injection, as orthotopic xenograft transplantation assay gave similar results with IMEC-MYC-PIK3CA[H1047R] forming highly proliferative and heterogeneous tumors expressing both luminal (CK8/18) and basal (CK5/6 and p63) markers (Fig. 6b, c). In addition, tumors were negative for ER-α and PR and did not show overexpression of HER2 (Fig. 6c), recapitulating the histo-pathological features of basal-like breast cancer. To determine long-term tumorigenic potential, xenograft-derived (XD) cells obtained from primary tumors were re-injected in the mammary gland of secondary recipient mice. Serial transplantations showed that the XD cells maintained tumorigenicity, forming tumors with features resembling the primary one (Fig. 6d). Notably, XD cells showed considerable migration and metastatic seeding capacity as, after

chirurgical resection of secondary tumors, all treated animals developed macro-metastasis in liver, lung, and spleen (Fig. 6e). Taken together, these results suggest that IMEC-MYC-PIK3CA[H1047R] were endowed with long-term tumorigenic capacity and metastatic potential.

**MYC-driven oncogenic signature in basal-like breast cancer.** We next asked whether the MYC-dependent oncogenic signature, activated in M2 by de novo enhancer modulation, could be associated with IMEC-MYC-PIK3CA[H1047R] tumorigenicity. We measured a significant transcriptional upregulation of these oncogenes in both primary and secondary tumors, in respect to parental cells (Fig. 7a and Supplementary Fig. 9g). Concomitantly, we determined activation of the respective enhancers in XD cells, suggesting that the same molecular mechanism driven by MYC in M2 cells was responsible for their upregulation in tumorigenic cells (Fig. 7b and Supplementary Fig. 9h). To assess whether our findings are clinically relevant, we investigated the expression of MYC-dependent oncogenic signature in a database of breast cancer patients. The average expression of MYC-induced oncogenes is strongly upregulated in basal-like breast cancers and predictive of a worst prognosis for this specific molecular subtype (Fig. 7c–e). Among the MYC targets, those that were specifically overrepresented within the basal-like breast cancers, which include modulators of kynurenine, prostaglandin, and Wnt pathways, are frequently deregulated in human cancers[60–62]. The pathological relevance of the identified MYC signature was further corroborated by the observation that the expression of these genes correlated with reduced metastatic-free survival in patients affected by high-grade breast cancer (Supplementary Fig. 9i). Taken together these data demonstrated that MYC-modulated enhancers activate oncogenic pathways, which are associated with basal-like breast cancer in patients with a poor prognosis.

## Discussion

In this work, we report the central role of MYC in initiating and sustaining a stepwise cell reprogramming process of mammary epithelial cells toward a stem cell-like condition, favoring tumor initiation and progression. Specifically, we show that MYC induces dedifferentiation toward a progenitor-like state achieved through downregulation of lineage-specifying TFs, resulting in decommissioning of luminal-specific enhancers. The oncogene-triggered loss of cell identity favors the acquisition of stem cell traits, which is mirrored by activation of de novo oncogenic enhancers. Importantly, the herein deciphered epigenetic reprogramming supports tumorigenesis as the oncogenic enhancers are reactivated in the transformed cell counterpart. We further show that MYC directly binds to a subset of de novo enhancers, suggesting that it participates in activating oncogenic pathways

---

**Fig. 5** Reactivation of WNT pathway supports MYC-induced stem cell features. **a** qRT-PCR of WNT pathway-related genes on IMEC WT, IMEC-MYC, and M2 clones #1 and #2, normalized on spike-in RNAs. Data are means ± SEM (n = 3) (*P < 0.05, **P < 0.01, ***P < 0.001; Student's t-test). **b** FACS analysis showing the GFP signal of IMEC-MYC-7TGP cultured in adhesion or as mammospheres. **c** On the left, phase contrast images showing IMEC-MYC-7TGP cultured in low adhesion conditions. Scale bar, 100 μm. On the right, FACS analysis showing GFP signal and dye retention profile of IMEC-MYC-7TGP cultured in low adhesion conditions. **d** Scheme representing GFP[high] and GFP[low] cells sorting from IMEC-MYC-7TGP, which gave rise to GFP[high]- and GFP[low]-derived 1° Spheres (M1). GFP[high]-derived 1° Spheres underwent a second single-cell sorting of GFP[high] and GFP[low] cells, which gave rise to GFP[high]- and GFP[low]-derived 2° Spheres (M2). Representative FACS analysis showing GFP signal and median fluorescence intensity (MFI) of sorted IMEC-MYC-7TGP and GFP[high]-derived M1 clones are reported. **e** On the left, single-cell spheres formation efficiency (SFE) of GFP[high] and GFP[low] cells sorted from IMEC-MYC-7TGP, which gave rise to M1 clones. MFI of GFP[high]- and GFP[low]-derived M1 clones is reported. On the right, single-cell SFE of GFP[high] and GFP[low] cells sorted from GFP[high]-derived M1, which gave rise to M2 clones. MFI of GFP[high]- and GFP[low]-derived M2 clones is reported. Data are means ± SEM (n = 4). **f** GSEA of mammary stem cells (MaSCs) gene signature in freshly sorted GFP[high] and GFP[low] cells (n = 3). **g** GSEA of lung, bone, and brain metastatic signatures in freshly sorted GFP[high] and GFP[low] cells (n = 3)

involved in the formation and maintenance of TICs. Overall, we established a key role of MYC as tumor reprogramming factor by guiding the acquisition of stem cell-like traits, thereby increasing the likelihood of neoplastic transformation upon further oncogenic insults.

In oncogenic setting, supraphyisiological activation of MYC promotes tumorigenesis by conferring selective cell growth advantage[63]. It has been proposed that high level of oncogenic MYC elicits RNA amplification by inducing a widespread increase of transcription elongation of the pre-existing active genes. Further increment of MYC levels in advanced tumorigenic cells determines chromatin invasion, which is characterized by MYC association with almost all the active promoters and enhancers[64,29,46]. However, the functional relevance of MYC occupancy at distal *cis*-regulatory elements in supporting tumorigenesis has not been fully elucidated. By investigating the functional relevance of MYC binding to enhancers, we found that MYC drives cellular reprogramming by inducing cell type-specific

enhancer decommissioning, combined with activation of oncogenic enhancers. Specifically, we showed that the MYC-driven transcriptional reprogramming is supported by repression of *cis*-regulatory elements controlling the expression of LDTFs. The finding that the master TFs GATA3 and ESR1 are repressed by MYC could explain the measured inactivation of luminal-related enhancers and the consequent downregulation of the luminal transcriptional program. Of importance, the MYC-repressed TFs are tumor suppressors whose downregulation is strongly associated with poorly differentiated basal-like breast cancers. Considering that in the same subset of breast cancers MYC pathway is frequently overactivated[25], our results suggest a possible functional link between transcriptional repression of these tumor suppressors and MYC activation in establishing and maintaining basal-like tumors. Further validation with primary luminal cells within an in vivo setting would be required to confirm that MYC overexpression causes transcriptional repression of GATA3 and

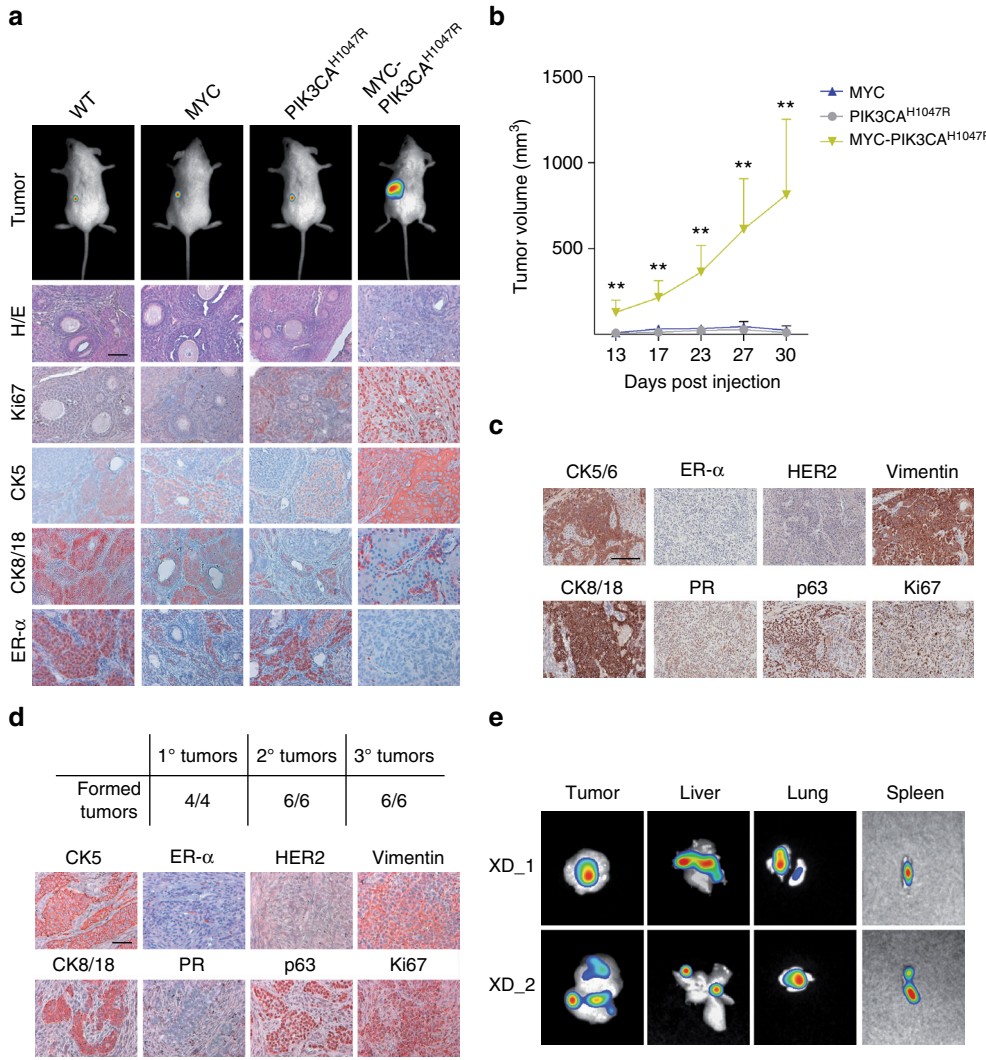

**Fig. 6** MYC-induced reprogramming favors the onset of TICs. **a** In the upper panel, whole body imaging of NOD/SCID mice 6 weeks after injection into the sub-renal capsule of indicated cells (*n* = 3). In the lower panel, representative hematoxylin & eosin (H/E) and immunohistochemical staining for indicated markers on generated xenografts. Scale bar, 50 μm. **b** Primary xenograft tumor volume (mm³) following orthotopic injection of indicated cells in NOD/SCID mice. Data are mean tumor size ± SEM (*n* = 4) (**P* < 0.01; Student's *t*-test; IMEC-MYC-PIK3CA[H1047R] compared with IMEC-MYC). **c** Representative immunohistochemical staining of indicated markers on primary tumors generated after orthotopic injection of IMEC-MYC-PIK3CA[H1047R] in NOD/SCID mice (*n* = 4). **d** In the upper panel, table depicting the number of formed tumors per injected mice in a serial transplantation assay. In the lower panel, representative immunohistochemical staining of indicated markers performed on tertiary tumors. Scale bar, 50 μm. **e** In vivo imaging of metastasis generated in NOD/SCID mice 4 weeks after secondary tumor removal. Representative images of two mice are shown (*n* = 6)

ESR1, thus triggering the acquisition of a progenitor-like state which could favor the formation of basal-like tumors.

The herein deciphered multistep reprogramming process consisted of the re-activation of a stem cell-like transcriptional program, mirrored by the establishment of a specific epigenetic landscape. We provide evidence that activation of de novo enhancers corresponds to increased expression of the associated genes, which are particularly enriched for pro-self-renewing TFs,

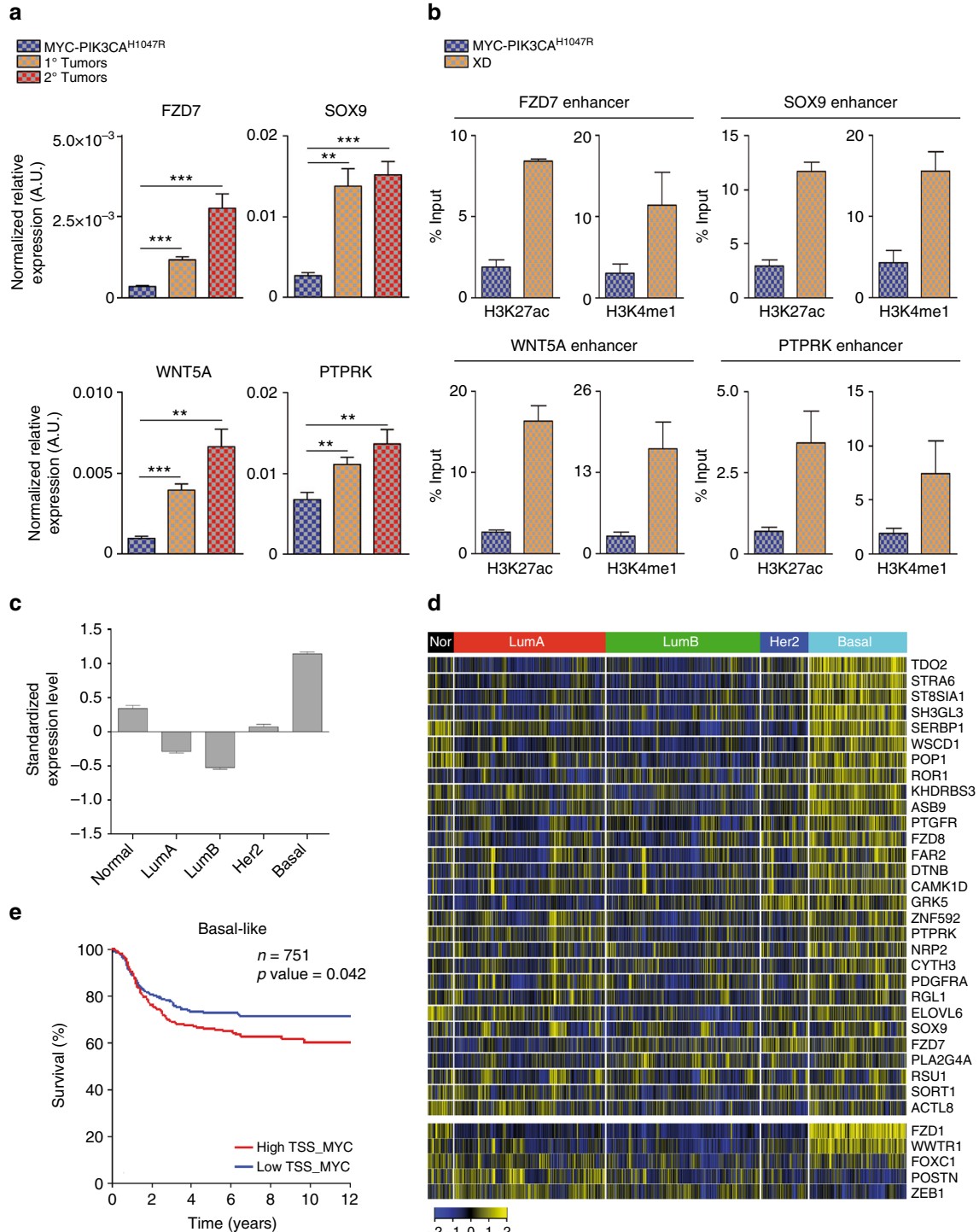

**Fig. 7** MYC-induced activation of oncogenic pathways in TICs. **a** qRT-PCR of genes associated to MYC-bound M2 de novo enhancers in IMEC-MYC-PIK3CA$^{H1047R}$ and 1° and 2° xenograft tumors cells, normalized on GAPDH. Data are means ± SEM ($n = 3$) (**$P < 0.01$, ***$P < 0.001$; Student's $t$-test). **b** ChIP-qPCR assessing H3K27ac and H3K4me1 deposition at the MYC-bound M2 de novo enhancer regions in IMEC-PIK3CA$^{H1047R}$ and XD cells. Data are means ± SEM ($n = 3$). **c** Average gene expression values of MYC direct target signature in breast cancer samples of the meta-data set stratified according to molecular subtype. Data are shown as mean ± standard error of the mean (SEM) ($P < 0.0001$; unpaired $t$test). **d** Gene expression levels of MYC direct target genes (upper heatmap) and of basal-specific enhancer associated genes (MYC un-bound; lower heatmap) in breast cancer samples of the meta-data set stratified according to molecular subtype. **e** Kaplan–Meier analysis representing the probability of metastasis-free survival in 751 basal-like breast cancer patients from the meta-data set stratified according to high or low MYC direct target signature score

with established roles in tumorigenesis. Moreover, components of both canonical and non-canonical Wnt signaling were being re-activated at this cellular state and we further demonstrate that Wnt activation represents a bona fide functional marker of MYC-induced reprogramming in mammary epithelial cells. Of importance, we showed that the same oncogenic enhancers are re-activated upon tumor initiation, causing overexpression of the regulated genes in both primary and secondary tumors. These findings suggest that combinatorial targeting of the hyper-activated oncogenic pathways may represent a therapeutic rationale to switch-off the MYC-dependent epigenetic repro-gramming, hampering its tumor initiation capacity.

Since MYC is not able to bind closed chromatin, the activation of de novo enhancers should be ascribed to the activity of pioneer TFs that induce a wave of chromatin remodeling that would allow MYC binding to open regions. Motif discovery analyses at enhancers associated with MYC binding showed enrichment for FOX- and SOX-family members, which have been previously shown to act as pioneer TFs. Given these observations, it could be interesting to verify whether FOXC1 and SOX9, which has been demonstrated to play a central role as regulator of breast cancers[65,66], could also act as pioneer factors in MYC-induced oncogenic cell reprogramming. Considering that their expression is modulated by the activation of the de novo enhancers, we argue that a self-reinforcing transcriptional network could support epigenetic reprogramming in TICs.

It has been suggested that oncogene-driven cell plasticity in which, following an appropriate oncogenic insult, most cell in a tissue has the potential to acquire stem cell-like properties, par-ticipates in determining tumor heterogeneity[5–7,10,11,67]. We pro-pose that resetting of the epigenetic landscape, which allows the establishment of a stem cell-like transcriptional program and predisposes cells to neoplastic transformation, should be therefore considered a hallmark of tumor initiation. This epigenetic remodeling can indeed cause a susceptible state, in which cells are more prone to acquire genetic alterations, going through trans-formation and tumor progression. With this work we identified MYC as a key player in oncogenic reprogramming as it triggers tumor initiation through the modulation of the epigenetic state of enhancers, thereby perturbing cell identity.

## Methods

**Cell lines**. All experiments were performed in the following cell lines and deri-vatives: hTERT-immortalized human mammary epithelial cells (IMEC, a kind gift from Dr. Micheal Cole), T47D, MCF7, and ZR751, retrieved from ATCC. These cell lines were used as representative of transformed mammary luminal epithelial cells. All cell lines were tested for mycoplasma contamination and resulted negative.

**Animal studies**. Animal studies were performed under the institutional guidelines of animal care committee, Italian Ministry of Health authorization (IACUC 373/2015-PR). Tumorigenicity studies were performed by the injection of $2 \times 10^6$ cells mixed 1:6 with Matrigel (BD Biosciences #354230) in 30 μl under the sub-renal capsule of 5-week-old NOD/SCID mice from Charles River Laboratories. Mice were killed after 4 weeks from injection. Limiting dilution experiments were per-formed by injecting $2 \times 10^5$, $2 \times 10^4$, $2 \times 10^3$, and $2 \times 10^2$ cells mixed 1:3 with Matrigel in 30 μl under the sub-renal capsule of NOD/SCID mice. Mice were killed after 12 weeks from injection. Serial transplantation experiments were performed by orthotopical injection of $2 \times 10^6$ cells, suspended in 30 μl of 1:6 Matrigel, in the mammary gland of 5-week-old NOD/SCID mice from Charles River Laboratories. Mice were killed when tumors reached $2 \times 2$ cm size. Tumor length and width was measured with electronic caliber and tumor size was calculated using the formula: (smaller diameter)$^2$ × larger diameter × π/6. For metastasis formation analysis, secondary tumors were removed and mice and organs were subjected to bioima-ging 4 weeks later. In order to allow in vivo tracking of tumor and metastasis formation, cells were transduced with a lentiviral vector encoding for Luciferase (pTween-Luc-NOeGFP) and animals were monitored after Luciferin (VivoGlo Luciferin, Promega) injection by using the Photon IMAGER (Biospace Lab). Data were analyzed with M3 Vision software. At the end of the experiments, mice were killed according to IACUC guidelines and tumors and metastasis collected for in vivo imaging, immunohistochemistry, RNA extraction, and cells isolation.

**Plasmids**. pMXs-c-Myc was a gift from Shinya Yamanaka (Addgene plasmid #13375); pBABE-puro-RAS V12 was a gift from Bob Weinberg (Addgene plasmid #1768); MSCV-p53DD-iGFP was a gift from Wechsel-Reya lab; PGK-H2BmCherry was a gift from Mark Mercola (Addgene plasmid # 21217); PIK3-CA$^{H1047R}$ was subcloned from pBabe-puro-HA-PIK3CA$^{H1047R}$, a gift from Jean Zhao (Addgene plasmid # 12524), into PGK-H2BmCherry; 7xTcf-eGFP//SV40-PuroR (7TGP) was a gift from Roel Nusse (Addgene plasmid #24305); pTween-Luc-NOeGFP was a gift from Stassi lab. Inducible IPTG-driven shRNAs for Myc and constitutive shRNAs for Myz-1 were purchased from Sigma-Aldrich.

**Cell culture**. hTERT-immortalized human mammary epithelial cells (IMEC) and XD cells were cultured at 37 °C and 5% CO$_2$ in 1:1 DMEM/F-12 medium (gibco #11320-074) supplemented with insulin (Clonetics, MEGM SingleQuots #CC-4136), EGF (Clonetics, MEGM SingleQuots #CC-4136), bovine pituitary extract (Clonetics, MEGM SingleQuots #CC-4136), hydrocortisone (Clonetics, MEGM SingleQuots #CC-4136) and 100 ng/ml cholera toxin (Sigma #8052). IMEC-MYC, IMEC-PIK3CA$^{H1047R}$, IMEC-P53DD, and IMEC-RAS were generated by trans-ducing IMEC with pMXs-c-Myc, PGK-PIK3CA$^{H1047R}$, pBABE-puro RAS V12, and MSCV-p53DD-iGFP vector, respectively. IMEC-MYC-7TGP cells were gen-erated by transduction of IMEC-MYC with 7TGP vector. T47D and MCF7 cells were cultured at 37 °C and 5% CO$_2$ in DMEM high glucose (Euroclone #ECB7501L) supplemented with 10% fetal bovine serum (Euroclone #ECS0180L), 1 mM sodium pyruvate (Euroclone #ECM0542D) and 2 mM glutamine (Euroclone #ECB3000D). ZR751 cells were cultured at 37 °C and 5% CO$_2$ in RMPI Medium 1640 (gibco #31870-025) supplemented with 10% fetal bovine serum, 2 mM glu-tamine and 1 mM sodium pyruvate. MCF7-MYC, T47D-MYC, and ZR751-MYC were generated by transduction with pMXs-c-Myc.

**Mammospheres culture and related assays**. Mammospheres culture was per-formed as previously described[41]. Briefly, single cells were plated in ultralow attachment plates (Corning) at a density of $2 \times 10^4$ viable cells/ml in 1:1 DMEM/F-12 medium supplemented with insulin (Clonetics, MEGM SingleQuots #CC-4136), EGF (Clonetics, MEGM SingleQuots #CC-4136), hydrocortisone (Clonetics, MEGM SingleQuots #CC-4136), B-27 Supplement (Gibco # 17504044), 20 ng/ml human FGF-basic (PeproTech #100-18B) and 5 μg/ml heparin (Sigma-Aldrich #H3149). Formed mammospheres were collected after 6 days. For long-term clo-nogenic assays, cells were transduced with PGK-H2BmCherry and single cells were plated in 96-well plates, in six technical replicates, at a density of $4 \times 10^3$ viable cells in 100 μl. After 6 days, fluorescence images of the entire wells were acquired. Then the cells were collected and passed in the same condition. This was repeated for four subsequent passages. Images were acquired with an Eclipse T$i$ fully automated system (Nikon); spheres formation efficiency (SFE) and mammospheres area (μm$^2$) were measured using the NIS Element software (Nikon). Objects with an area <2000 μm$^2$ (diameter < 50 μm) were excluded from the analysis. Single-cell clo-nogenic assay was performed in 96-well plates, in at least three biological replicates. Single cells were sorted with a BD FACS Aria III sorter (BD Biosciences), one cell/well and formed mammospheres were counted after 3 weeks by microscope observation (time window required for primary spheres formation).

**Immunofluorescence**. For mammospheres differentiation assay, cells were grown in mammospheres culture conditions for 6 days, then mammospheres were col-lected and left lay down on collagen I-coated glass coverslips, in mammospheres medium supplemented with 10% FBS. After 7 days mammospheres were fixed for 20 min at room temperature with 4% paraformaldehyde (Sigma-Aldrich #158127). Coverslips were processed for immunofluorescence according to the following conditions: permeabilization and blocking with PBS/1% BSA/0.3% Triton X-100 (blocking solution) for 1 h at room temperature, followed by incubation with primary antibody (diluted in the blocking solution) for 2 h at RT, three washes in the blocking solution and incubation with secondary antibodies (diluted in the blocking solution) for 30 min at room temperature.

Images were acquired using a Leica TCS SP5 confocal microscope with HCX PL APO ×63/1.40 objective. Confocal z stacks were acquired with sections of 0.35 μm. In cases where image analysis was performed, image acquisition settings were kept constant.

Primary antibodies are as follows: CK8 (Covance #1E8-MMS-162P), CK14 (Covance #AF64-155P), ER-α (Merk Millipore #F3-A 04-1564), α-SMA (abcam #ab5694). Cell nuclei were visualized with DAPI (Sigma). Secondary antibodies were goat-anti-mouse or -rabbit coupled to Alexa-488 or -568 (Invitrogen).

**Flow cytometry analysis (FACS)**. ALDH activity was assessed with the Aldefluor kit (Stemcell Technologies #1700) on IMEC WT and IMEC-MYC cultured as mammospheres for one passage.

Dye retention assay was performed with CellTrace Violet Cell Proliferation Kit (molecular probes #C34557) on IMEC-MYC-7TGP cultured as mammospheres for one passage. After the staining, cells were re-plated in the same conditions and acquired to FACS after 6 days.

**Tumor digests**. Tumors were chopped into small pieces in sterile conditions, then incubated at 37 °C for 2 h in DMEM/F12 containing 2% bovine serum albumin,

300 U/ml collagenase III (Worthington #M3D14157) and 100 U/ml hyaluronidase (Worthington #P2E13472). Following digestion, tumor cell suspensions were pelleted and then suspended in 0.25% trypsin for 2 min.

**Immunohistochemical analysis.** Immunohistochemical analyses were carried out on 5-μm-thick paraffin-embedded sections of breast cancer xenografts. Tissues underwent antigen retrieval, where required permeabilized with cold 0.1% Triton X 100, and stained overnight at 4 °C with antibodies against Ki67 (MIB 1, Dako), p63 (4A4, Santa Cruz), CK5 (XM26, Novocastra), CK 5/6 (D5/16B4, Dako), CK8/18 (5D3, Novocastra), Vimentin (R28, Dako), PR (16, Biocare Medical), Her2 (D8F12, CST), and ER (6F11, Novocastra). Successively, sections were incubated with biotinylated secondary antibodies and exposed to streptavidin-peroxidase (Dako). Stainings were revealed with 3-amino-9-ethylcarbazole substrate (AEC, Dako) and nuclei counterstained with aqueous hematoxylin. For H/E, tissues were stained with hematoxylin for 2 min and subsequently with eosin for 1 min.

**Soft agar assay.** Colony-forming assay was carried out using Noble agar (Sigma-Aldrich #A5431). For the lower layer, agar was mixed with IMEC medium, reaching a final concentration of 0.6% and plated on six-well plates. $4 \times 10^5$ cells were plated on top of it, in 0.3% agar. Colony formation was monitored up to 21 days by microscope observation.

**Invasion assay.** $2 \times 10^3$ cells were plated in medium without growing factors and placed onto Matrigel-coated (BD Biosciences #354230) transwells of 8-μm pore size (Corning #3422). In the lower part of the transwell DMEM supplemented with 10% AB human serum (Euroclone, ECS0219D) was placed as a chemo-attractant. The number of migrated cells in the lower chamber was calculated up to 72 h by microscope observation.

**Protein extraction and western blot analysis.** Total protein extracts were obtained as follows. Cells were washed twice with cold PBS, harvested by scrapping in 1 ml cold PBS and centrifuged for 5 min at 1500 rpm. Harvested cell pellets were lysed by the addition of 5× v/v ice-cold F-buffer 30 min at 4 °C. The chromosomal binding proteins were then separated using BioRuptor waterbath sonicator (Diagenode) at low setting for 5 min. Samples were sonicated in pulse of 30 s with 30 s intervals. Lysates were cleared by centrifugation for 10 min at 14,000 rpm at 4 °C and supernatant was collected on ice. Protein concentration of lysates was determined using Pierce™ BCA Protein Assay Kit 24 (Thermo Scientific, 574 #23227), according to the manufacturer's instructions. The absorbance was measured at $\lambda = 595$ using SAFAS spectrophotometer (SAFAS, Monaco). Values were compared to a standard curve obtained from the BSA dilution series.

For western blots analysis, 20 μg of protein samples were boiled and loaded onto a pre-cast Bolt 4−12% Bis-Tris Plus gels (Novex #NW04122BOX) and run in Bolt MES running buffer (Novex #B0002). After electrophoresis, proteins were transferred to a nitrocellulose membrane. Membranes were blocked in PBS-T containing 5% Blotting-Grade Blocker (BIO-RAD #170-6404) (blocking buffer), for 1 h at RT with constant agitation and incubated with indicated primary antibody O/N at 4 °C with agitation. The membrane was then washed three times with PBS-T, each time for 5 min, followed by incubation with secondary antibody HRP-conjugated for 1 h at RT. ECL reagents (GE Healthcare #RPN2232) was used to initiate the chemiluminescence of HRP. The chemiluminescent signal was captured using LAS3000 system (GE Healthcare).

Primary antibodies used are as follows: β-Actin (Sigma-Aldrich #A5441), c-Myc (Cell Signaling #5605). Relative optical density was quantified with ImageJ Software. Uncropped images of all the western blots are reported in Supplementary Fig. 10.

**RNA extraction, expression level quantification and microarray experiments.** Total RNAs were extracted from log-phase cells with TRIzol (Ambion #15596018), according to the manufacturer's instructions. Quantitative real-time PCR analysis was performed with SuperScript III One-Step SYBR Green kit (Invitrogen #11746). Relative gene expression levels were determined using comparative Ct method, normalizing data on endogenous GAPDH or ERCC RNA Spike-In Mix (Ambion #4456704). Primers used in this study are listed in Supplementary Table 4.

For microarray experiments, 500 ng of each sample of RNA were processed to generate labeled cRNAs following the Illumina TotalPrep RNA amplification Kit (Ambion #AMIL1791) protocol. cRNA concentration was quantified and subjected to quality control on Agilent Bioanalyzer (Agilent Technologies #554 G2943CA) and hybridized to HumanHT-12 v4 BeadChip Arrays (Illumina #15011977).

**Microarray analysis.** BeadChip Arrays were scanned with HiScan Array Scanner (Illumina) using the iScan Control Software (Illumina). Genes and probes transcript levels were obtained from Illumina Intensity Data (.idat) files, applying quantile normalization and background subtraction implemented by the GenomeStudio Gene Expression Module v1.0 Software (Illumina). All experiments in each condition reported were performed on triplicate biological samples. Signals associated with a $p$-value > 0.05 in all samples were discarded from the analysis.

Cut-off for up- and downregulation of gene expression was set to twofold change threshold in all the analyses performed.

**Computational analysis of gene expression data.** Scatter plots, correlation heatmaps and PCA analysis of gene expression data were performed in R (http://www.R-project.org/). Differentially expressed genes were checked for biological and functional enrichment using the GO-based online tool PANTHER Classification System. Geneset Enrichment Analysis (GSEA) was performed with genesets retrieved from both public available databases and indicated papers (Supplementary Data 1).

**Chromatin immunoprecipitation (ChIP) assay.** Each ChIP experiment was performed in at least three independent biological samples. Briefly, cells were crosslinked with 1% formaldehyde for 10 min at RT and the reaction was quenched by glycine at a final concentration of 0.125 M, for 5 min at RT. Cells were lysed in lysis buffer (50 mM Tris-HCl pH 8, 0.1% SDS, 10 mM EDTA pH 8, 1 mM phenylmethyl sulphonyl fluoride (PMSF, Sigma # P7626), protease inhibitor cocktail (Roche #04693159001)) and chromatin was sonicated to an average size of 0.1–0.5 kb, using a Branson D250 sonifier (four cycles of 30 s, 20% amplitude). Fifty micrograms of each sonicated chromatin was incubated O/N at 4 °C with 4 μg of indicated antibodies (anti-MYC sc-764 Santa Cruz Biotechnology; anti-trimethyl histone H3 Lys4 07-473 Millipore; anti-monomethyl histone H3 Lys4 8895 Abcam; anti-acetyl histone H3 Lys27 4729 Abcam; anti-MIZ-1 10E2[37,38]). Protein G-coupled Dynabeads (Thermo Fisher Scientific # 10004D) were blocked O/N at 4 °C with 1 mg/ml sonicated salmon sperm DNA (Thermo Fisher Scientific #AM9680) and 1 mg/ml$^{-1}$ BSA. Subsequently, blocked protein G-coupled Dynabeads were added to the ChIP reactions and incubated for 4 h at 4 °C. Dynabeads linked to ChIP reactions were then recovered and resuspended in RIPA buffer (10 mM Tris-HCl, pH 8, 0.1% SDS, 1 mM EDTA, pH 8, 140 mM NaCl, 1% DOC, 1% Triton, 1 mM PMSF, protease inhibitor cocktail). Magnetic beads were sequentially washed five times with ice-cold RIPA buffer, twice with ice-cold RIPA-500 buffer (10 mM Tris-HCl, pH 8, 0.1% SDS, 1 mM EDTA, pH 8, 500 mM NaCl, 1% DOC, 1% Triton, 1 mM PMSF, protease inhibitor cocktail), twice with ice-cold LiCl buffer (10 mM Tris-HCl, pH 8, 0.1% SDS, 1 mM EDTA, pH 8, 250 mM LiCl, 0.5% DOC, 0.5% NP-40, 1 mM PMSF, protease inhibitor cocktail) and once with TE buffer (10 mM Tris-HCl, pH 8, 1 mM EDTA, pH 8, 1 mM PMSF, protease inhibitor cocktail). Crosslinking was then reversed in direct elution buffer (10 mM Tris-HCl, pH 8, 0.5% SDS, 5 mM EDTA, pH 8, 300 mM NaCl) at 65 °C O/N. Finally, DNA was purified using Agencourt AMPure XP SPRI beads (Beckman #A63882), washed twice in EtOH 70% and dissolved in 60 ml of Tris-HCl, pH 8.0. DNA was analyzed by quantitative real-time PCR using SYBR GreenER master mix (Thermo Fisher Scientific # 11762500). All experimental values were shown as percentage of input. To take into account background signals, we subtracted the values obtained with a non-immune serum to the relative ChIP signals (anti-mouse IgG CS200621 Millipore).

**Antibodies.** All the antibodies used in this study and the working dilutions for all the experimental settings are reported in Supplementary Table 5.

**ChIP-seq library generation and data analysis.** Five nanograms of immunoprecipitated and purified DNA was used to generate ChIP-seq libraries. Briefly, end repair of DNA fragments was achieved by sequential 15 min incubations at 12 °C and 25 °C with 0.15 U/μl T4 PNK (NEB #M0201L), 0.04 U/μl T4 POL (NEB #M0203L) and 0.1 mM dNTPs (NEB #N0446S). A-base addition was performed by incubating end-repaired DNA fragments with 0.25 U/μl of Klenow fragment (NEB # M0212L) and 167 μM dATP (NEB N0440S) for 30 min at 30 °C. Adaptor ligation was achieved by using the Quick ligation kit (NEB #M2200L) and performing an incubation of 15 min at 25 °C. Processed DNA fragments were finally amplified with a thermal cycler for 14 cycles, by using the PfuUltra II Fusion HS DNA Pol kit (Agilent #600674). All DNA purification steps between the different enzymatic reactions were performed using Agencourt AMPure XP SPRI beads (Beckman #A63882). The obtained libraries were subjected to quality control on Agilent Bioanalyzer (Agilent Technologies #G2943CA) before sequencing them with Illumina HiSeq2000. Sequenced reads were aligned to the human genome (GRCh37/hg19) by using Bowtie2 version 2.2.3 and only uniquely mapped reads were used in the subsequent analyses. In order to find the regions of ChIP-seq enrichment over background, we used different peak callers. For MYC ChIP-seq we used MACS2 ($p$-value < $1 \times 10^{-6}$), while for histone modifications we used SICER V1.1 (window size = 200; gap size = 200; FDR < 0.01). The HOMER software command "getDifferentialPeaks" was used to find ChIP-seq differentially enriched regions between different IMEC samples (cut-offs = twofold change and $p$-value $1 \times 10^{-4}$). The HOMER software command "annotatePeaks.pl" was used to assign peaks and enhancer regions to the nearest genes, according to GRCh37/hg19 annotation, and to count the number of tags from different sequencing experiments on those regions. Tag counts were then used to produce heatmaps with TM4 MeV v4.9 software. Annotated genes were checked for biological and functional enrichment using both the GO-based online tool PANTHER and GSEA, with genesets retrieved from both publicly available databases and indicated papers. Venn diagrams were generated using the online tool. Tag density plots around the center of enhancers regions were quantified with the ngsplot 2.47 command ngs.

plot.r and plotted with GraphPad Prism (GraphPad Software, San Diego, California, USA, www.graphpad.com). Normalized BigWig tracks of ChIP-seq experiments were generated with bedtools 2.24.0 and the bedGraphToBigWig program and visualized in the UCSC Genome Browser. Motifs enrichment analysis was performed with the online tool Analysis of Motif Enrichment (AME) of the MEME suite v4.11.2 and with the HOMER command "findMotifsGenome.pl". In all the analyses reported data are normalized by per million mapped reads (RPM). For Miz-1 global binding analysis (Supplementary Fig. 5e-j), data were retrieved from GEO: GSE44672[37] and GSE59146[38]. ChIP-seq data were mapped to human genome (GRCh37/hg19) by using Bowtie2 and the HOMER software and ngsplot 2.47 was used to count normalized reads to generate box-plots and tag density plots, respectively. For transcriptional profile analysis, normalized data from microarray experiments were directly retrieved from GSE59145[38], while RNA-seq data retrieved from GSE44672[37] were aligned to human genome (GRCh37/hg19) by using Bowtie2 and normalized read counts were obtained by using the htseq-count tool from HTSeq and DESeq2.

**Collection and processing of breast cancer gene expression data**. Breast cancer gene expression data have been obtained from a collection of 4640 samples from 27 major data sets comprising microarray data of breast tumors annotated with pathological information and clinical outcome (Supplementary Table 2). All data were measured on Affymetrix arrays and have been downloaded from NCBI Gene Expression Omnibus (GEO, http://www.ncbi.nlm.nih.gov/geo/) and EMBL-EBI ArrayExpress (http://www.ebi.ac.uk/ arrayexpress/). Prior to analysis, all data sets have been re-organized as described in ref. [68]. Since raw data (.CEL files) were available for all samples, integration, normalization, and quantification of gene expression levels have been obtained with the procedure described in ref. [69]. The type and content of pathological and clinical annotations have been standardized, among the various data sets, as described in ref. [49]. This resulted in a compendium (meta-data set) comprising unique samples from 25 independent cohorts (Supplementary Table 3). Breast cancer data of the METABRIC collection, comprising microarray gene expression profiles and clinical annotations for 997 cancer samples, were downloaded from the European Genome-Phenome Archive (EGA, http://www.ebi. ac.uk/ega/) under accession number EGAD00010000210[39]. Original Illumina probe identifiers were mapped to Entrez gene IDs using the Bioconductor illuminaHumanv3.db annotation package for Illumina HT-12 v3 arrays obtaining log2 intensity values for a total of 19,761 genes. The 522 primary tumors of the TCGA breast cancer data set (The Cancer Genome Atlas Network, 2012) were downloaded from the Comprehensive molecular portraits of human breast tumors portal (Level 3 data arcive; https://tcga-data.nci.nih.gov/docs/publications/brca_2012/).

Breast cancer molecular subtypes have been assigned using the PAM50 intrinsic subtype classifier, i.e., the intrinsic.cluster.predict function, of the genefu R package. Gene expression heatmaps have been generated using the function heatmap.2 of R gplots package after row-wise standardization of the expression values. Average signature expression has been calculated as the standardized average expression of all signature genes in sample subgroups (e.g., molecular subtypes). The values shown in graphs are thus adimensional. Signature scores have been obtained summarizing the standardized expression levels of signature genes into a combined score with zero mean[70]. All gene expression analyses were performed in R (version 3.2.0).

**Survival analysis**. To identify two groups of tumors with either high or low MYC direct target signature, we used the classifier described in ref. [70], that is, a classification rule based on the MYC direct target signature score. Briefly, tumors were classified as MYC direct target signature Low if the signature score was negative and as MYC direct target signature High if the signature score was positive. To evaluate the prognostic value of the signature, we estimated, using the Kaplan–Meier method, the probabilities that patients would remain free of metastasis. The Kaplan–Meier curves were compared using the log-rank (Mantel–Cox) test. $p$-values were calculated according to the standard normal asymptotic distribution. Survival analysis was performed in GraphPad Prism.

**Quantification and statistical analysis**. Statistical parameters are reported in figure legends and include: number of replicates analyzed ($n$), dispersion and precision measures (mean ± SEM) and statistical significance ($p$-value). Data have been statistically assessed by one-tailed Student's $t$-test and indicated in figure legends. In figures, asterisks mean $*p < 0.05$, $**p < 0.01$, $***p < 0.001$, ns = not significant. $p < 0.05$ and lower were considered significant. All experiments were performed in at least triplicate biological replicates. For microarray analysis, transcript levels were quantile normalized and background subtracted using the GenomeStudio Gene Expression Module v1.0 Software (Illumina). Signals associated with a $p$-value > 0.05 in all samples were discarded from further analysis and gene expression values <1 were set to 1. The threshold to define up- and down-regulated genes was set to twofold changes. For ChIP-seq analysis the following statistical parameters were used: MACS2 peak calling of MYC ChIP-seq ($p$-value < $1\times10^{-6}$); SICER V1.1 peak calling of histone modification (FDR < 0.01); HOMER "getDifferentialPeaks" (cut-offs = twofold change and $p$-value < $1\times10^{-4}$). All ChIP-seq data are normalized by library depth and reported as per million mapped reads (RPM). Statistical $p$-value of GO, GSEA and AME are reported in figures. Data collection and analyses of all studies involving animals were conducted randomly and not blinding.

**Data availability**. Data resources: Raw and quantile normalized data files for the microarray analysis have been deposited in the NCBI Gene Expression Omnibus under accession number GSE86407.

Raw data and genomic regions of ChIP-seq peaks have been deposited in the NCBI Gene Expression Omnibus under accession number GSE86412.

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

## Acknowledgements

We thank Monica Moro and Mariacristina Crosti for their help with FACS analyses; Francesco Ferrari and Mattia Forcato for their suggestions on ChIP-seq data analyses. We thank Martin Eilers for providing the anti-MIZ-1 10E2 antibody. We are grateful to Stefano Biffo, and Stefano Piccolo for insightful discussion and critical reading of the manuscript. Work in the Bicciato's lab was supported by AIRC Special Program Molecular Clinical Oncology "5 per mille" (Grant n. 10016) and by Epigenomics Flagship Project (EPIGEN). Work Todaro's lab was supported by AIRC (Grant n. 14415); A Turdo is recipient of an AIRC fellowship; A. Chiinici is a PhD student of the molecular and experimental medicine PhD programme at University of Palermo in partnership with Humanitas University. Work in Zippo's group was supported by grants from the Italian health ministry (GR-2010-2319033; GR-2011-02351172), by Epigenomics Flagship Project (EPIGEN) and CARIPLO foundation (2014-0915); V. Poli is recipient of an AIRC fellowship (21158).

## Author contributions

V.P., L.F., A.C., S.M., A.F. and A.Z. conceived the study, designed the experiments and interpreted the data. A.M. provided essential reagents and expertise. G.G., V.V., A.T., M.G., A.C., E.L., and M.T. performed the in vivo experiments. V.P., L.F., A.C., S.M., S.F., and A.F. performed the cellular and molecular biology studies. L.F. and S.B. performed computational data analysis. S.Bo. reviewed critically the work. M.T. designed the in vivo experiments and interpreted the data. A.Z. supervised the work and wrote the manuscript.

## Additional information

**Competing interests:** The authors declare no competing interests.

