## [Peer Review File · Nature Communications]

Reviewers' comments:

Reviewer #1 (Remarks to the Author):

Review Poli

The manuscript addresses the role of MYC in the behavior of luminal epithelial breast cells and corresponding tumor cells. Introducing MYC into such cells, for the non-transformed model TERT-immortalized cells (IMEC) were used, results in the downregulation of mature luminal genes and the activation of genes associated with luminal progenitor cells, resulting in the dedifferentiation of the cells to a progenitor-like state. This resulted in cells with an increased potential to form mammospheres over several passages, indicative of enhanced long-term self-renewal capacity. Interestingly only MYC, but not other oncogenes, could induce this effect. These findings and studies with a clonal cell (M2) reveal that MYC induces a stem cell-like trait. An important finding is that under the experimental conditions many promoters with H3K4me3 peaks are not occupied by MYC, suggesting that it is not a broad, general activation of promoters due to strong overexpression of MYC, which has resulted in some confusion in the past. Moreover, enhancers that are down-regulated seem to lose MYC, unlike what is shown for ESR1, and are enriched in motifs for factors that are luminal specific. A distal set of enhancers gains MYC and is activated in the M2 population. The MYC cells show an activation of the WNT signaling pathway with a rather broad spectrum of signal intensities. The WNT high cells are more enriched for stem cell-like properties. Interestingly the IMEC-MYC cells could be transformed by the expression of an activated form of PIK3CA. The changes in gene expression in the M2 cells are comparable to those seen in the IMEC-MYC-PIK3CA cells compared to the controls. Finally these changes seem to be relevant in tumor samples.

Specific comments:

A major concern is that for all the experiments a single M2 clone is used. Establishing cell clones can result in the selection of highly specific lines that may not represent the overall effects. This heterogeneity is also documented when the WNT reporter cells are analyzed (in MYC cells). The high and low GFP-expressing cells have distinct properties. Depending on the origin of the M2 cells, although like from the WNT high population, quite distinct findings might be expected. Thus, this is a limitation of the study which otherwise is very interesting and thorough.

Fig. 1f: Two master regulators, ESR1 and GATA3, which are down-regulated in response to MYC are analyzed. This correlation is supported by analyzing transcript and protein levels from public resources. Enhanced binding of MYC to an enhancer and the promoter, respectively, resulted in a reduction of positive histone marks. This remains unexplained and is in need of a mechanism. Is MYC directly causing this change in histone marks?

The ESR1 enhancer shown in Fig. 1 is also not typical. When compared to the findings shown in Fig. 3, the impression is that enhanced MYC binding correlates with activation of enhancers. This should be clarified.

Fig. 1g: Genes associated with ML enhancer signature are down-regulated by MYC. What does this signature mean, which factors are relevant, and is MYC binding to all of them? Is the pattern seen for the enhancer of ESR1 prototypical?

Fig. 3e shows de novo enhancers. Many of those seem to be bound by MYC in the M2 cells. The further analysis in Fig. 4c only uses a subset of these genes. Does this mean that only roughly 900 enhancers are associated with genes that are induced, i.e. the majority of the 5800 enhancers are not linked to genes that are induced?

Reviewer #2 (Remarks to the Author):

The manuscript by Poli et al revisits the role of MYC in breast cancer tumorigenesis using hTERT-immortalized mammary epithelial cells (IMEC) in which MYC is overexpressed at moderate levels (IMEC-Myc). Moreover, they generated single cell derived mammospheres (M2) from the latter and used them for analysis along the other two cell systems. The authors propose that MYC represses lineage-specific transcription by affecting the associated enhancers. In addition, novel enhancers are activated that in toto results in the formation of TICs and increased metastasis formation. While the concept, that MYC overexpression is preventing differentiation while promoting self-renewal is not new and simply the way MYC functions, the authors claim that this effect is happening at the level of enhancers.

The paper relies almost entirely on in vitro culture systems (there is some limited cross validation in patient cohorts at the end) with a questionable origin of the M2 cells. Some of the bioinformatics studies appear preliminary and in part seem rather weak. However, as I am not an expert in this area, these parts need to be looked at by a knowledgeable bioinformatics expert. Altogether, the study is interesting, although the data fall somewhat short to the conclusions made by the authors.

Main Critique:

(1) The authors base a lot of data on the single cell derived mammospheres (M2). In fact, the most striking differences are seen not in the IMEC vs. IMEC-Myc comparison but between the latter and the M2 (Fig. 3-5). As MYC has not been changed in M2, the dramatic molecular and cellular changes are likely not a consequence of MYC, but rather the way these M2 cells are growing as mammospheres. Moreover, as they are derived from single cells, are the M2s used in the paper all derived from one clone, or from many clones, or from different clones in each experiment? Have the authors checked whether the used M2 cells have acquired additional mutations? Myc promotes genetic instability and it is likely that the M2 after prolonged culture are genetically changing and certain variants may be selected (genetic drift). If yes, do some of the additional genetic lesions lead to further up-regulation of MYC?

(2) Fig. 4A: a correlation of $r=0.168$ is very weak and would actually suggest that there is no significant correlation.

(3) For all the gene set enrichment analysis (GSEA) the authors show the enrichment score (ES). This is unusual, typically the normalized enrichment score (NES) is shown as this corrects for multiple testing (FDR).

According to the Broad institute: The ES must be adjusted to account for differences in the gene set sizes and in correlations between gene sets and the expression data set. The resulting normalized enrichment scores (NES) allows to compare the analysis results across gene sets" (<http://software.broadinstitute.org/cancer/software/genepattern/modules/docs/GSEA/14>). As mentioned above a bioinformatics expert should be consulted.

(4) The introduction of PIK3CA mutant (Fig. 6) followed by an in vivo analysis seems detached from the rest of the study. Here the effect of PIK3CA on top of MYC overexpression is analysed, which does not really address Myc but rather PIK3CA function.

Additional points to address:

(a) Fig. 1d: normalized relative expression → to what is it normalized to?

(b) Fig. 1F: only one small promoter region each was selected – why this one and not any of the other active regions – conclusions cannot be drawn from short single elements...

(c) Fig. 1g: ML enhancer signature: what is the origin of this?

Reviewers' comments: Reviewer #1 (Remarks to the Author):

The manuscript addresses the role of MYC in the behavior of luminal epithelial breast cells are corresponding tumor cells. Introducing MYC into such cells, for the non-transformed model TERT-immortalized cells (IMEC) were used, results in the downregulation of mature luminal genes and the activation of genes associated with luminal progenitor cells, resulting in the dedifferentiation of the cells to a progenitor-like state. This resulted in cells with an increased potential to form mammospheres over several passages, indicative of enhanced long-term self-renewal capacity. Interestingly only MYC, but not other oncogenes, could induce this effect. These findings and studies with a clonal cell (M2) reveal that MYC induces a stem cell-like trait. An important finding is that under the experimental conditions many promoters with H3K4me3 peaks are not occupied by MYC, suggesting that it is not a broad, general activation of promoters due to strong overexpression of MYC, which has resulted in some confusion in the past. Moreover, enhancers that are down-regulated seem to lose MYC, unlike what is shown for ESR1, and are enriched in motifs for factors that are luminal specific. A distal set of enhancers gains MYC and is activated in the M2 population. The MYC cells show an activation of the WNT signaling pathway with a rather broad spectrum of signal intensities. The WNT high cells are more enriched for stem cell-like properties. Interestingly the IMEC-MYC cells could be transformed by the expression of an activated form of PIK3CA. The changes in gene expression in the M2 cells are comparable to those seen in the IMEC-MYC-PIK3CA cells compared to the controls. Finally these changes seem to be relevant in tumor samples.

Specific comments: A major concern is that for all the experiments a single M2 clone is used. Establishing cell clones can result in the selection of highly specific lines that may not represent the overall effects. This heterogeneity is also documented when the WNT reporter cells are analyzed (in MYC cells). The high and low GFP-expressing cells have distinct properties. Depending on the origin of the M2 cells, although like form the WNT high population, quite distinct findings might be expected. Thus, this is a limitation of the study which otherwise is very interesting and thorough.

We are pleased that Reviewer#1 found the major findings of work very interesting. However, He/She was concerning about the possible limitation of using a single clone of mammospheres (M2), which could result in an over-interpretation of the obtained results.

We agree with Reviewer#1 about the necessity of showing that the major conclusions of this work are not dependent on clonal selection. Indeed, many of the presented data showing the MYC-induced clonogenic potential have been already performed by using independent-derived clones of mammospheres. Specifically, in the original manuscript the experiments in which we measured the single cell clonogenic capacity and the long-term maintenance (Fig. 2f and Fig. 5d-e of the revised manuscript) resulted from the analyses of four independent primary spheres and their derivatives (secondary spheres) after sub-cloning. We reported the aggregated analyses in Fig. 2f and the individual clonogenic capacity of each clone in Supplementary Fig. 3g of the revised manuscript. Similarly, we reported the results obtained by analyzing the clonogenic potential of GFP+ vs GFP- cells both as aggregated and individual data in Fig. 5d-e and Supplementary Fig. 8c, respectively. Furthermore, we confirmed that the enrichment in the mammospheres of cells endowed with self-renewing capacity was dependent on the exogenous MYC as its knock-down was sufficient to impaired the clonogenic potential of these mammospheres, as reported in Supplementary Fig. 3h-i of the revised manuscript.

Beside the functional assays, we investigated whether the defined activation of a stem cell-like transcriptional and epigenetic program could be also recapitulated in other independent clones of mammospheres. We assessed the gene expression profiling of three new independent clones of secondary mammospheres (named M2#2, M2#3 and M2#4) and performed comparative analyses

with the M2 clone (named M2#1), whose gene expression pattern has been described in the original manuscript. Both PCA and Pearson Correlation clearly showed that the independent clones of mammospheres activated a common transcriptional program, which clearly distinguished them from IMEC wt and IMEC-MYC (shown in the Supplementary Fig. 4a-b). Moreover, scatter plot analyses showed that most of the genes expressed by the previously analyzed M2 clone are in common with the newly profiled independent clones of mammospheres, as showed by Supplementary Fig. 4c. In addition, we established that those genes involved in sustaining self-renewing capacity resulted induced also in independent clones (M2#2-4), respect to IMEC-MYC (shown in the Supplementary Fig. 4d-e), confirming the re-activation of a pluripotency-associated transcriptional program. To further confirm that the changes of gene expression in mammospheres were not related to clonal selection, we also measured in an independent clone the altered transcript levels of the Wnt related genes (Fig. 5a) and MYC-targets (shown in the Supplementary Fig. 7b-c). These comparative analyses showed that independently from the clonal origin, both set of genes were regulated at the same extend in the mammospheres. Notably, the expression of the same genes resulted dependent on the exogenous MYC as its knock-down impaired their modulation (shown in the Supplementary Fig. 8a). Furthermore the activation of the *de novo* enhancers, which supported the induction of MYC-targets was also confirmed in the analyzed independent clone (shown in the Supplementary Fig. 7e-f), arguing in favor of the establishment of an alternative epigenetic program. Together, the additional analyses support the notion that the observed phenotype and the activation of both transcriptional and epigenetic program did not result from clonal selection but rather were driven by MYC activation.

Fig. 1f: Two master regulators, ESR1 and GATA3, which are down-regulated in response to MYC are analyzed. This correlation is supported by analyzing transcript and protein levels form public resources. Enhanced binding of MYC to an enhancer and the promoter, respectively, resulted in a reduction of positive histone marks. This remains unexplained and is in need of a mechanism. Is MYC directly causing this change in histone marks?

We further investigated the mechanism through which MYC induced transcriptional repression of the two master regulators ESR1 and GATA3. We found that MYC directly repressed their expression by forming a complex with MIZ1, antagonizing its pro-transcriptional activity, as shown in Fig. 1g of the revised manuscript. We also showed that the ESR1 and GATA3 down-regulation relied on MIZ1 as its knock-down impaired their MYC-dependent transcriptional repression (shown in the Supplementary Fig. 2f-g). Reciprocally, by knocking-down the exogenous MYC with two IPTG-inducible shRNAs we rescued the transcript level of these two TFs (shown in the Supplementary Fig. 2d-e). Considering that these analyses highlighted that the increment of MYC binding at cis-regulatory elements induced transcriptional repression on the analyzed genes, we investigated whether this may represent a common pattern driven by MYC activation. By analyzing the MYC occupancy genome-wide we defined two subset of genes, which resulted been induced or repressed in response to the increased MYC binding (show in Fig. 3b-c of the revised manuscript). The down-regulated genes showed a lower level of MYC occupancy in the steady state (IMEC wild-type), which was moderately increased upon MYC activation. The opposite trend was instead measured among the up-regulated targets, with a higher MYC occupancy that was related with the enrichment for E-boxes (show in Fig. 3b-c and Supplementary Fig. 5a-d). Of importance, when we analyzed the MYC and MIZ1 binding on the same subset of genes, using data retrieved by independent CHIP-seq analyses, we confirmed the described pattern (as shown in Supplementary Fig. S5e-i). Of relevance, the MYC-repressed genes were also MIZ1 targets in both the analyzed datasets, suggesting that the MYC-MIZ1 interplay plays a relevant role in dictating the transcriptional outcome for their targets. Overall the genome-wide profiling for TFs occupancy highlighted that not only MYC associated with gene activation but also with

transcriptional repression, independently from the analyzed cellular system. These analyses thus support the notion that among others, cell lineage transcription factors could be directly repressed by MYC, which antagonizes the transcriptional activity of MIZ1.

The ESR1 enhancer shown in Fig. 1 is also not typical. When compared to the findings shown in Fig. 3, the impression is that enhanced MYC binding correlates with activation of enhancers. This should be clarified.

We agree with the Reviewer#1 that MYC-dependent ESR1 enhancer modulation differs from the luminal-specific enhancers, which resulted decommissioned in response to MYC activation. In the revised manuscript we clarified that ESR1 enhancer was regulated similarly to a large subset of MYC targets in which the increment of MYC occupancy on the promoters caused transcriptional repression (shown in Fig. 3b-c). Of note, we also showed that these genes are commonly bound by MYC and MIZ1, (as retrieved by analyzing independent ChIP-seq datasets) suggesting that MYC binding antagonized the transcriptional activity of MIZ1 thus causing transcriptional repression. Of importance we also showed in the revised manuscript that MIZ1/MYC complex co-bound also the ESR1 enhancer indicating that the same competition-based mechanism could determine the down-regulation of ESR1 (Shown in Fig. 1g of the revised manuscript).

Fig. 1g: Genes associated with ML enhancer signature are down-regulated by MYC. What does this signature mean, which factors are relevant, and is MYC binding to all of them? Is the pattern seen for the enhancer of ESR1 prototypical?

We apologize for not clearly stated in the manuscript how we defined this signature and we corrected it in the revised version of the manuscript. The ML enhancer signature was retrieved by identifying those enhancers that were co-bound by luminal-specific TFs including GATA3, ESR1, FOXA1 and ZNF217¹. In this work we showed that luminal-specific TFs were down-regulated in response to MYC activation (shown in Fig. 1d) and we reasoned that their downstream effectors, whose regulation relies on enhancer activation, would consequently being inactivated. The GSEA analysis showed that indeed the associated genes were down-modulated in response to MYC activation (shown in Fig. 1e in the revised manuscript). Of note, by analyzing our ChIP-seq data, we could not measure a significant enrichment of MYC association on these enhancers (data not shown), indicating that their modulation is not a direct consequence of MYC binding but rather it could depend on altered expression of luminal-specific TFs. Said that, further experiments determining the changes of occupancy of the luminal-specific TFs would be required to address this point. However this aspect does not represent the focus on this work and would not be included in the revised version of the manuscript.

Fig. 3e shows de novo enhancers. Many of those seem to be bound by MYC in the M2 cells. The further analysis in Fig. 4c only uses a subset of these genes. Does this mean that only roughly 900 enhancers are associated with genes that are induced, i.e. the majority of the 5800 enhancers are not linked to genes that are induced?

In Fig 3e (corresponding to Fig. 3f in the revised manuscript) we showed *de novo* enhancers and the relative enrichment of MYC binding in M2. In the original manuscript, the following analysis in Fig 4a was focused on the top 100 enhancer-associated genes that resulted over-expressed in M2 respect to IMEC-MYC. Of note, by defining the differentially expressed genes and the relative enrichments of MYC binding using a 2-fold cut-off, we adopted stringent criteria to limit the analyses on those targets (top 100) whose expression was highly induced in mammospheres. Among those we further confirmed that they were truly target of MYC and their up-regulation was consequent to *de novo* enhancer activation (as shown in Supplementary Fig. 7b-d). In the revised manuscript we repeated the correlation analyses between MYC binding increment and gene expression changes in which we

included all the differentially expressed genes (keeping the 2-fold change as threshold), as shown in Fig. 4a. The obtained results further support the notion that MYC recruitment on the *de novo* enhancers correlated with changes in gene expression ($r^2=0.68$). This represents a specific feature related to MYC binding on the *de novo* enhancers, as previous analyses could not retrieve a direct association between MYC occupancy on enhancer and gene expression ².

Regarding the relative numbers of genes whose expression would change in response to *de novo* enhancer activation, we would like to underline that the selected parameters to associate enhancer to gene activation do not permit to fully address this point. Specifically using a criterion of proximity to assign each enhancer to its regulated gene, we impose that each enhancer would be functionally associated to a single gene, which is located within 100kb from the enhancer. This represents an approximation of all the possible interactions between enhancers and promoters, given the 3D organization of the chromatin. However, although this criterion reduces the number of enhancer-promoters associations, it permits to identify a subset of genes whose expression is modulated in response to enhancer activation. In addition, the arbitrary threshold selected to define those enhancers in which we uncertainly measured and increment of MYC occupancy combined with induction of gene expression (> 2-fold), reduced the numbers of identified genes whose expression changed in response to *de novo* enhancers activation. Therefore we could not exclude that using other selective criteria or performing 3C chromatin conformation analyses will not permit to retrieve a higher number of associated genes that are regulated by the *de novo* enhancers. However, we believe that these further analyses are not the main scope of this paper in which instead we defined for the first time that MYC-driven activation of *de novo* enhancers establishes an alternative transcriptional program which support self-renewal potential and it is further activated in tumorigenic cells *in vivo*.

Reviewer #2 (Remarks to the Author):

The manuscript by Poli et al revisits the role of MYC in breast cancer tumorigenesis using hTERT-immortalized mammary epithelial cells (IMEC) in which MYC is overexpressed at moderate levels (IMEC-Myc). Moreover, they generated single cell derived mammospheres (M2) from the latter and used them for analysis along the other two cell systems. The authors propose that MYC represses lineage-specific transcription by affecting the associated enhancers. In addition, novel enhancers are activated that in toto results in the formation of TICs and increased metastasis formation.

While the concept, that MYC overexpression is preventing differentiation while promoting self-renewal is not new and simply the way MYC functions, the authors claim that this effect is happening at the level of enhancers.

The paper relies almost entirely on in vitro culture systems (there is some limited cross validation in patient cohorts at the end) with a questionable origin of the M2 cells. Some of the bioinformatics studies appear preliminary and in part seem rather weak. However, as I am not an expert in this area, these parts need to be looked at by a knowledgeable bioinformatics expert. Altogether, the study is interesting, although the data fall somewhat short to the conclusions made by the authors.

Main Critique:

(1) The authors base a lot of data on the single cell derived mammospheres (M2). In fact, the most striking differences are seen not in the IMEC vs. IMEC-Myc comparison but between the latter and the M2 (Fig. 3-5). As MYC has not been changed in M2, the dramatic molecular and cellular changes are likely not a consequence of MYC, but rather the way these M2 cells are growing as mammospheres. Moreover, as they are derived from single cells, are the M2s used in the paper all derived from one clone, or from many

clones, or from different clones in each experiment? Have the authors checked whether the used M2 cells have acquired additional mutations? Myc promotes genetic instability and it is likely that the M2 after prolonged culture are genetically changing and certain variants may be selected (genetic drift). If yes, do some of the additional genetic lesions lead to further up-regulation of MYC?

We agree with Reviewer#2 that we focused our attention on the differences occurring between M2 and IMEC-MYC as we were interested in understanding the contribution of the defined *de novo* enhancers in regulating gene expression and also in supporting tumorigenesis. Indeed we showed that the same enhancers get re-activated in xenograft-derived tumor cells driving the transcriptional activation of the related genes. The Reviewer#2 is concerning about the robustness of the described regulatory mechanisms as the obtained data relies on the analyses performed on a single M2 clone of mammospheres. We acknowledge that this may represent a limitation and in the revised manuscript we addressed this point extensively.

1. The functional assays in which we measured the contribution of MYC in supporting the enrichment of stem cell-like cells were originally performed using different clones of mammospheres in each experiment. We now showed the aggregated analyses of the clonogenic assays in Fig. 2f and Fig. 5d-e, while the individual analyses performed for each clone are shown in Supplementary Fig. 3g and Supplementary Fig. 8c, respectively. All the other experiments in which we measured sphere-forming capacity and differentiation potential were performed by using bulk population of mammospheres.
2. We performed gene expression profiling of three additional clones (named M2#2, M2#3 and M2#4), which were independently derived by single cell cloning. We performed comparative analyses with the M2 clone (named M2#1), by PCA and Pearson Correlation which clearly showed that the independent clones shared a common transcriptional program, respect to IMEC wt and IMEC-MYC (shown in the Supplementary Fig. 4a-b). Moreover, scatter plot analyses demonstrate that the majority of expressed genes in M2 (previously analyzed in the revised manuscript) are in common with the three independent clones (shown the Supplementary Fig. 4c). Of importance, these analyses also highlighted genes involved in supporting self-renewing capacity were also up-regulated in the new clones (M2#2-4), as shown in Supplementary Fig. 4d-e. Furthermore, to determine the contribution of the cell culture condition to the activation of the stem cell-like transcriptional program, we compared by PCA the gene expression pattern of IMEC cells grown in the mammospheres medium. As shown it is evident that the cell culture conditions were not sufficient to activate the same gene expression pattern retrieved by the independent clones of mammospheres (see attached picture). In addition, if the maintenance of cells as mammospheres would be sufficient to direct the functional, transcriptional and epigenetic changes detected in this work, then it would be expected that these cells could be maintained as mammospheres independently from the activation of MYC. Said that, we extensively demonstrated that IMEC wt could not be propagated as mammospheres and the expression of other oncogenes did not support the long-term maintenance of mammospheres to the same extent as MYC overexpression did (shown in Fig. 2a-e and Supplementary 3a-f). Finally we extended the qRT-PCR analyses on individual genes to a second clone of mammospheres and comparative analyses showed that these genes were regulated with the same pattern in the two independent clones of mammospheres (shown in Fig. 5a and in the Supplementary Fig. 7b-c). Furthermore, the activation of the *de novo* enhancers, which supported the induction of MYC-targets was also confirmed in the analyzed independent clone (shown in the Supplementary Fig. 7e-f), arguing in favor of the establishment of an alternative epigenetic program. These data exclude that the

activation of the stem cell-like transcriptional program is somehow influenced by clonal selection.

Supporting Figure showing PCA analysis of IMEC WT, MYC and four independent clones of mammospheres. The gene expression profiling of IMEC WT and IMEC-MYC was performed using both standard growth factor medium (maintenance in adhesion) and mammospheres medium (sphere medium).

3. We also checked whether the prolonged maintenance of MYC-sustained mammospheres could cause genetic alterations that could explain the measured functional and molecular changes that favored the onset of tumorigenesis *in vivo*. By performing FACS analyses we did not detect aneuploidy and the karyotype resulted normal, as shown in Supplementary Fig. 4g-h. We also showed that MYC protein level did not change in four independent clones of primary mammospheres and their derivatives upon sub-cloning (secondary and tertiary mammospheres; Supplementary Fig. 4f). These results argue against the possibility that genetic alterations could perturb the protein level of MYC and consequently transcriptional and epigenetic program. Of importance, we also checked whether additive genetic alterations combined with MYC over-expression could activate a pro-tumorigenic program. Specifically we performed orthotopic transplantation assays with the M2 mammospheres and measured the formation of primary tumors in the injected mice. Although we used the same methodology described in the original paper for the IMEC-MYC-PIK3CA^{H1047R}, we never observed tumor growth and after six months the mice resulted healthy with no sign of local tissue damage or hyper-proliferative cells. These data have not been included in the revised manuscript but in our opinion are relevant for addressing this issue. Finally we showed in the revised manuscript that the acute knockdown of the exogenous MYC in the established clones impaired the self-renewal capacity of the mammospheres (Supplementary Fig. 3h-i), thus indicating a reversible phenotype that is poorly compatible with genetic drift and clonal selection.

(2) Fig. 4A: a correlation of $r=0.168$ is very weak and would actually suggest that there is no significant correlation.

In this analysis we measured the correlation between MYC enrichment at the *de novo* enhancers and increment of gene expression among the top 100 differentially expressed targets. We focus on this subset of genes as we were interested in defining the most responsive genes to the recruitment of MYC on these *de novo* enhancers to then measure their expression among breast cancer patients, dissecting their clinical relevance (as shown in Fig. 7c-e). In the revised manuscript we instead measure the same correlation between MYC recruitment and gene expression, but we included all the enhancer-associated genes, which were induced in M2 respect to IMEC-MYC (> 2-fold increment). From a

technical point of view, to take into account the measured variability in gene expression level we performed the correlation analyses using a bin of 10 genes. Using these criteria we measured a stronger correlation between increment of MYC occupancy and changes in gene expression ($r^2=0.68$), as shown in Fig. 4a of the revised manuscript. These data, together with the association between the increment of MYC occupancy at the *de novo* enhancers and the relative changes in gene expression (Fig. 4d) indicated that MYC binding contributed to the transcriptional activation of the related genes.

(3) For all the gene set enrichment analysis (GSEA) the authors show the enrichment score (ES). This is unusual, typically the normalized enrichment score (NES) is shown as this corrects for multiple testing (FDR). According to the Broad institute: The ES must be adjusted to account for differences in the gene set sizes and in correlations between gene sets and the expression data set. The resulting normalized enrichment scores (NES) allows to compare the analysis results across gene sets“ (<http://software.broadinstitute.org/cancer/software/genepattern/modules/docs/GSEA/14>). As mentioned above a bioinformatics expert should be consulted.

We appreciate that Reviewer #2 accurately revised our GSEA results and we agree with her/his consideration that “typically the normalized enrichment score (NES) is shown as this corrects for multiple testing (FDR)”. In fact, GSEA is typically used to assess entire databases of gene sets and the NES, associated to relative FDR, is required in this kind of analysis. This is clearly stated both in the original paper presenting the method (Subramanian A. et al., PNAS, 2005) and also in the Broad institute guidelines.

However, in our analysis we never run GSEA on either databases of gene sets or even more than 3 gene sets. We rather use a very limited number of gene sets per analysis, generated from data retrieved from literature. Therefore, we are far away from the “*more than 30*” data sets suggested and we do not aim to “*compare analysis results across gene sets*” but rather to evaluate the enrichment score of a given data set between two samples. In this cases we cannot use the NES (with associated FDR), but it is correct to show the ES with relative p-values, which fail to account for multiple hypothesis testing (that’s to say multiple data sets), but are statistically significant for single gene sets, as clearly stated in both the original paper presenting the method and also in the Broad institute guidelines.

In conclusion, following the Broad institute guidelines we are confident that it is correct to show ES with relative p-values as this represents the most correct way to evaluate the statistical significance of our analysis comprising individual gene sets, in comparison with the NES with associated FDR, which are, instead, required to compare and evaluate the significance of a large number of gene sets.

For clarity we here report the statement from both the original paper presenting the method (Subramanian A. et al., PNAS, 2005) and also in the Broad institute guidelines referring when is appropriate use the NES and the ES, respectively;

NES: “*Adjustment for Multiple Hypothesis Testing. When an entire database of gene sets is evaluated, we adjust the estimated significance level to account for multiple hypothesis testing. We first normalize the ES for each gene set to account for the size of the set, yielding a normalized enrichment score (NES). We then control the proportion of false positives by calculating the false discovery rate (FDR) corresponding to each NES*”

and also in the Broad institute guidelines (<http://software.broadinstitute.org/cancer/software/genepattern/modules/docs/GSEA/14>) and (<http://software.broadinstitute.org/gsea/doc/GSEAUUserGuideFrame.html>):

“*Typically, GSEA is run with a large number of gene sets. For example, the MSigDB collection and subcollections each contain hundreds to thousands of gene sets. This has implications when comparing enrichment results for the many sets: the ES must be adjusted to account for differences in the gene set*

sizes and in correlations between gene sets and the expression data set. The resulting normalized enrichment scores (NES) allow you to compare the analysis results across gene sets. The nominal p-values need to be corrected to adjust for multiple hypothesis testing. For a large number of sets (rule of thumb: more than 30), we recommend paying attention to the False Discovery Rate (FDR) q-values: consider a set significantly enriched if its NES has an FDR q-value below 0.25”.

“The normalized enrichment score (NES) is the primary statistic for examining gene set enrichment results. By normalizing the enrichment score, GSEA accounts for differences in gene set size and in correlations between gene sets and the expression dataset; therefore, the normalized enrichment scores (NES) can be used to compare analysis results across gene sets. The false discovery rate (FDR) is the estimated probability that a gene set with a given NES represents a false positive finding”.

ES: “Step 1: Calculation of an Enrichment Score. We calculate an enrichment score (ES) that reflects the degree to which a set S is overrepresented at the extremes (top or bottom) of the entire ranked list L.”

“Step 2: Estimation of Significance Level of ES. We estimate the statistical significance (nominal P value) of the ES by using an empirical phenotype-based permutation test procedure that preserves the complex correlation structure of the gene expression data.”

(<https://software.broadinstitute.org/cancer/software/gsea/wiki>

[/index.php/FAQ](https://software.broadinstitute.org/cancer/software/gsea/wiki/index.php/FAQ) and <http://software.broadinstitute.org/gsea/doc/GSEAUUserGuideFrame.html>):

“The nominal p value estimates the statistical significance of the enrichment score for a single gene set. However, when you are evaluating multiple gene sets, you must correct for gene set size and multiple hypothesis testing. Because the p value is not adjusted for either, it is of limited value when comparing gene sets.”

“The nominal p value estimates the significance of the observed enrichment score for a single gene set. However, when you are evaluating multiple gene sets, you must correct for multiple hypothesis testing. The FDR is the estimated probability that a gene set with a given enrichment score (normalized for gene set size) represents a false positive finding.”

(4) The introduction of PIK3CA mutant (Fig. 6) followed by an *in vivo* analysis seems detached from the rest of the study. Here the effect of PIK3CA on top of MYC overexpression is analysed, which does not really address Myc but rather PIK3CA function.

In a comparative analysis, we measure the contribution of different genetic hints in supporting long-term self-renewal capacity of mammospheres. Apart from MYC, none of the tested RAS, PIK3CA mutant and P53 was able to maintain prolonged self-renewal capacity indicating that the MYC-induced enrichment of stem cell like cells was not merely linked to cell transformation. However, the MYC overexpression itself was not sufficient to induce tumorigenesis *in vivo*, both in mammary gland and in the more susceptible environment such as under the renal capsule of immune-deficient mice. On the basis of these results we reasoned that a second genetic hint would be necessary to trigger cell transformation and by performing *in vitro* transformation assays we found that combining MYC with PIK3CA^{H1047R} was sufficient to induce anchorage independent cell growth (as shown in Supplementary Fig. 8a-d of the revised manuscript). Considering that we were focusing on defining oncogenic signaling that could synergize with MYC, we focused on those pathways that are recurrently deregulated in basal-like breast cancer. Among others, PIK3CA signaling pathway is frequently deregulated in TNBC and its targeting is currently under clinical investigation for treating these patients. Respect to the contribution of PIK3CA mutant, we demonstrated that the over-activation of PIK3CA pathway was not sufficient to induce enrichment of stem cell-like cells (shown in Supplementary Fig. 3f and Supplementary Fig. 9d) nor anchorage-independent cell growth (shown in Supplementary Fig. 9a-c) or tumor growth *in vivo* (shown in Fig. 6a-b). In the performed experiments we instead measure the contribution of MYC-driven activation of stem cell-transcriptional program to

tumor initiation and maintenance. In this work we showed how MYC-induced reprogramming of mammary luminal cells towards basal/stem cell-like state occurred independently from cell transformation and it favored the onset of tumorigenesis. Furthermore, we showed that the induced transcriptional program is maintained in tumorigenic cells through the activation of the herein identified *de novo* enhancers. In this regard our data indeed measured the role of MYC to sustain cell reprogramming and its contribution for the onset of tumorigenesis and the long-term maintenance of cancer cells with metastatic potential.

Additional points to address:

(a) Fig. 1d: normalized relative expression → to what is it normalized to?

As stated in the figure legend, the Fig 1d (Fig. 1f in the revised manuscript) represents qRT-PCR of GATA3 and ESR1 transcripts in IMEC WT and IMEC-MYC, whose levels were normalized on synthetic spike-in RNAs.

(b) Fig. 1F: only one small promoter region each was selected – why this one and not any of the other active regions – conclusions cannot be drawn from short single elements...

As for any ChIP assays each amplicon can cover a small region of the genome. In this case, within the promoter of GATA3 we focused on the genomic regions in which we retrieved by ChIP-seq the peak of MYC binding, confirming its association together with MIZ1. Regarding the histone modifications, we agree that a single amplicon could not be informative about the overall changes of a cis-regulatory elements and for these reasons we performed the ChIP-seq to detect the same histone modifications. The performed ChIP-assays were designed to confirm the altered patterns that we detected by performing the ChIP-seq, validating the robustness of our genome-wide analyses.

(c) Fig. 1g: ML enhancer signature: what is the origin of this?

We apologize for not clearly state in the manuscript how we defined this signature and we corrected it in the revised version of the manuscript. The ML enhancer signature was retrieved by identifying the genes associated to those enhancers that were co-bound by GATA3, ESR1, FOXA1 and ZNF217, in MCF7 cells ¹. In this work we showed that luminal-specific TFs were down-regulated in response to MYC activation (shown in Fig. 1d) and the GSEA analysis showed that indeed the enhancer-associated genes (herein defined as ML enhancer signature) were down-modulated in response to MYC activation (shown in Fig. 1e in the revised manuscript).

Supporting References

- 1 Frietze, S. *et al.* Global analysis of ZNF217 chromatin occupancy in the breast cancer cell genome reveals an association with ERalpha. *BMC Genomics* **15**, 520, doi:10.1186/1471-2164-15-520 (2014).
- 2 Walz, S. *et al.* Activation and repression by oncogenic MYC shape tumour-specific gene expression profiles. *Nature* **511**, 483-487, doi:10.1038/nature13473 (2014).

Reviewer #1 (Remarks to the Author):

The authors have adequately addressed all my concerns. The other reviewer suggested that the statistics should be evaluated by an expert reviewer. I agree with this as I am also not a specialist. There is certainly sufficient statistical data analysis and processing to justify that a biostatistician evaluates the work.

Minor comments:

Introduction page 1: ... into normal > in normal; oncogenic-driven > oncogene-driven

Page 2: evidences indicated > evidence indicates; Fagnocchi reference needs to be formatted.

Fig. 2a: should TTS read TSS?

Results page 2: the sentence between Ref 32 and 33 needs re-writing.

Page 4: ... in respect to ... (also elsewhere in the text)

Page 6: The statement from "Comparative analysis Importantly, ..." is not clear and would profit from re-writing.

Page 7: caused > cause

Page 8: ... which resulted unmarked Unclear what this means.

Discussion page 3: hypeactivated > hyperactivated; "... Myc is not able to bind closed chromatin ..."
This statement needs to be referenced. How clear is this? Could Myc be a pioneering factor for these enhancers and cooperate with the mentioned factors?

Reviewer #2 (Remarks to the Author):

The authors have thoroughly revised their manuscript and addressed all the points and issues raised in my review.

Reviewer #3 (Remarks to the Author):

The statistical analyses in this study are generally fine.

1) Figure 3h. X-axis is missing

2) Figure 6b. Two-way ANOVA: which two factors were included?

3) Details for GSEA are missing: how to get the gene rank list, which model was used (classic model vs. weighted model) and which/how many genes signatures were included.

Reviewers' comments:

Reviewer #1 (Remarks to the Author): The authors have adequately addressed all my concerns. The other reviewer suggested that the statistics should be evaluated by an expert reviewer. I agree with this as I am also not a specialist. There is certainly sufficient statistical data analysis and processing to justify that a biostatistician evaluates the work.

We are delighted that the Reviewer #2 acknowledges that we addressed all the issues in the revised version of the manuscript.

*Minor comments: Introduction page 1: ... into normal > in normal; oncogenic-driven > oncogene-driven
Page 2: evidences indicated > evidence indicates; Fagnocchi reference needs to be formatted. Fig. 2a: should TTS read TSS?*

We corrected the typos, as suggested.

Results page 2: the sentence between Ref 32 and 33 needs re-writing.

We changed the sentence as follows: "Importantly, genes whose expression is dependent on luminal-specific TFs binding on their cognate enhancers resulted down-regulated in IMEC-MYC."

Page 4: ... in respect to ... (also elsewhere in the text)

We corrected the typos, as suggested.

Page 6: The statement from "Comparative analysis Importantly, ..." is not clear and would profit from re-writing. Page 7: caused > cause

We corrected the typos, as suggested.

Page 8: ... which resulted unmarked Unclear what this means. Discussion page 3: hypeactivated > hyperactivated; "... Myc is not able to bind closed chromatin ..." This statement needs to be referenced. How clear is this? Could Myc be a pioneering factor for these enhancers and cooperate with the mentioned factors?

We changed the sentence as follows: "which did not carry...". We corrected the typos, as suggested. We also add the requested reference regarding the inability of MYC in binding closed chromatin.

Reviewer #2 (Remarks to the Author):

The authors have thoroughly revised their manuscript and addressed all the points and issues raised in my review.

We are delighted that the Reviewer #2 acknowledges that we addressed all the issues in the revised version of the manuscript.

Reviewer #3 (Remarks to the Author):

The statistical analyses in this study are generally fine.

We are delighted that the Reviewer #3 acknowledges the quality of the statistical analyses performed in this work.

1) *Figure 3h. X-axis is missing*

We corrected the figure, as suggested.

2) *Figure 6b. Two-way ANOVA: which two factors were included?*

The two factors are the timing and the volumes for each injected cell line.

3) *Details for GSEA are missing: how to get the gene rank list, which model was used (classic model vs. weighted model) and which/how many genes signatures were included.*

We included the requested additional information about the gene lists. All the analyses were performed using the weighted model. The number of gene signature analyzed simultaneously ranged from 1 to 4, in accordance with figure panels disposition.

Reviewers' comments:

Reviewer #3 (Remarks to the Author):

2) Figure 6b. Two-way ANOVA: which two factors were included?

The two factors are the timing and the volumes for each injected cell line.

This is not correct. Tumor volume is the response variable in the analysis and can not be the factor in the two-way ANOVA.

The two factors should be timing and the treatment effect.

But if so, there should be only one significant level for the whole tumor growth curve, rather than different significant levels marked at different time points in the figure.

That is why I raised this point. It is likely that the authors actually did t-test or one-way ANOVA at different time points to get the significant level (*, ***).

Please clarify this.

Reviewers' comments:

Reviewer #3 (Remarks to the Author):

2) Figure 6b. Two-way ANOVA: which two factors were included? The two factors are the timing and the volumes for each injected cell line. This is not correct. Tumor volume is the response variable in the analysis and can not be the factor in the two-way ANOVA. The two factors should be timing and the treatment effect. But if so, there should be only one significant level for the whole tumor growth curve, rather than different significant levels marked at different time points in the figure. That is why I raised this point. It is likely that the authors actually did t-test or one-way ANOVA at different time points to get the significant level (, ***). Please clarify this.*

We agree with Reviewer #3 that we did not apply the correct statistical analysis to assess our data and we apologize for this. We therefore revised the analysis in line with the current literature by performing student's t-test and modify the figure and the text accordingly, to clarify this point. In this respect, we specifically determined the significance of the changes in the tumor volumes occurring at the different time point by comparing the IMEC-MYC with the IMEC-MYC-PIK3CA^{H1047R}.

REVIEWERS' COMMENTS:

Reviewer #3 (Remarks to the Author):

the authors have addressed my comment.